# Governing Equation Discovery from Data Based on Differential Invariants

## Abstract

The explicit governing equation is one of the simplest and most intuitive forms for characterizing physical laws. However, directly discovering partial differential equations (PDEs) from data poses significant challenges, primarily in determining relevant terms from a vast search space. Symmetry, as a crucial prior knowledge in scientific fields, has been widely applied in tasks such as designing equivariant networks and guiding neural PDE solvers. In this paper, we propose a pipeline for governing equation discovery based on differential invariants, which can losslessly reduce the search space of existing equation discovery methods while strictly adhering to symmetry. Specifically, we compute the set of differential invariants corresponding to the infinitesimal generators of the symmetry group and select them as the relevant terms for equation discovery. Taking DI-SINDy (SINDy based on Differential Invariants) as an example, we demonstrate that its success rate and accuracy in PDE discovery surpass those of other symmetry-informed governing equation discovery methods across a series of PDEs. Additional results further indicate that our method exhibits strong robustness to data and symmetry noise, as well as significant potential for solving high-dimensional dynamic systems.

## 1 Introduction

Explicit equations, particularly partial differential equations (PDEs), play a significant role in scientific fields due to their concise and intuitive mathematical forms. Discovering governing equations directly from observational data has become an important topic, and its solutions may serve as AI assistants to human scientists in uncovering new physical laws. Although neural PDE solvers also aim for data-driven evolution prediction (Greydanus et al., 2019; Bar-Sinai et al., 2019; Sanchez-Gonzalez et al., 2020; Li et al., 2020; Thuerey et al., 2021; Brandstetter et al., 2022b; Gupta & Brandstetter, 2022; Takamoto et al., 2022; 2023; Lippe et al., 2023; Kapoor et al., 2023; Cho et al., 2024; Musekamp et al., 2024), their implicit learning approach, compared to explicit equation discovery, suffers from limitations such as lack of interpretability and weaker out-of-distribution (OOD) generalization. In this paper, we formalize the problem as discovering the governing PDE $F(x, u^{(n)}) = 0$ from trajectory data $u(x)$, where $x \in \mathbb{R}^p$ represents the independent variables, $u \in \mathbb{R}^q$ denotes the dependent variables, and $u^{(n)}$ signifies derivatives of $u$ with respect to $x$ up to order $n$.

Some previous works have made progress on the data-driven equation discovery problem. One category of search-based methods (Schmidt & Lipson, 2009; Gaucel et al., 2014; Petersen et al., 2019; Cranmer et al., 2019; 2020; Udrescu & Tegmark, 2020; La Cava et al., 2021; Mundhenk et al., 2021; Sun et al., 2022; Cranmer, 2023) explores the structure of equations interpretably, but their enormous search space incurs high computational costs. Another category of deep learning-based approaches (Brunton et al., 2016; Champion et al., 2019; Biggio et al., 2021; Messenger & Bortz, 2021; Kamienny et al., 2022) is generally more efficient and versatile, yet still requires pre-specifying key relevant terms of the equation skeleton. To address the limitations of these works, we need to leverage prior knowledge of scientific problems to constrain the form of equations—in other words, to narrow the search space of equations.

Symmetry is important prior knowledge in scientific problems, with each symmetry corresponding to a conserved quantity. Recently, some studies have attempted to discover symmetries from data for symmetry-dependent downstream tasks (Benton et al., 2020; Dehmamy et al., 2021; Moskalev et al., 2022; Desai et al., 2022; Yang et al., 2023; 2024a; Ko et al., 2024; Shaw et al., 2024; Hu et al.,

2025a). Our goal is to leverage known symmetries to guide the discovery of governing equations. Although Yang et al. (2024b) achieve this by adding explicit symmetry constraints or implicit symmetry regularization terms, the governing equations they identify cannot strictly adhere to general symmetries, and the manually specified equation skeletons significantly affect accuracy.

In this paper, we implement symmetry-guided equation discovery based on differential invariants. Given the infinitesimal generators of a symmetry group, we can derive their prolongation forms and differential invariants. Then, we directly select these differential invariants as the relevant terms and plug them into any existing equation discovery method, such as SINDy (Brunton et al., 2016). The proposition cited in Section 4.2 will demonstrate that this approach hard-embeds symmetry into the equation skeleton without sacrificing its expressive power. In other words, we "losslessly" compress the search space of equations. As shown in Figure 1, for the relatively complex nKdV equation $e^{-\frac{t}{t_0}} u_t + u u_x + u_{xxx} = 0$, existing equation discovery methods struggle to identify the key relevant terms and construct the correct equation skeleton from a large search space of partial derivatives, whereas our method can accurately determine it by leveraging the information of the symmetry group.

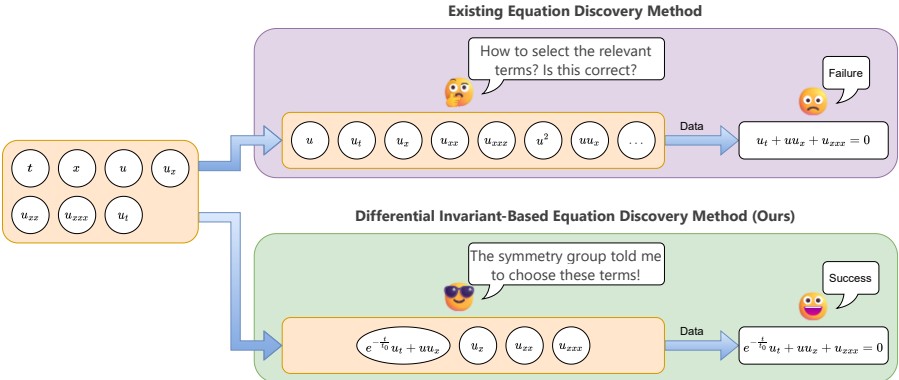

Figure 1: Comparison between the existing equation discovery method and our differential invariant-based equation discovery method for the nKdV equation $e^{-\frac{t}{t_0}} u_t + u u_x + u_{xxx} = 0$. The former struggles with selecting relevant terms, whereas our relevant terms are directly determined by the symmetry group.

In summary, our contributions are as follows: (1) we propose a method for equation discovery based on differential invariants, which is guided by symmetry groups in the selection of key relevant terms; (2) using the existing proposition, we substantiate that our method ensures the equation skeleton strictly adheres to symmetry without compromising its expressive power; (3) taking SINDy based on Differential Invariants (DI-SINDy) as an example, we demonstrate that our method can be plug-and-play with existing equation discovery approaches; (4) the experimental results on a series of PDEs show that our DI-SINDy achieves higher success rates and accuracy compared with baseline methods, while also exhibiting greater stability in long-term predictions.

## 2 RELATED WORK

**Symmetry discovery.**  The application of symmetry in downstream tasks is based on the premise that we know it in advance; otherwise, we first need to discover the symmetry from the data. Some works discover symmetry based on Lie group and Lie algebra representations (Dehmamy et al., 2021; Moskalev et al., 2022; Desai et al., 2022; Yang et al., 2023; Hu et al., 2025b), but they are limited to linear symmetries. Subsequent works attempt to find more complex nonlinear symmetries (Yang et al., 2024a; Ko et al., 2024; Shaw et al., 2024; Hu et al., 2025a). They utilize the discovered symmetries to guide downstream tasks, achieving performance improvements, which validates the effectiveness of the results. The techniques in this paper can be combined with these symmetry discovery methods to address scenarios where symmetries are not known in advance.

**Governing equation discovery.** Automatically discovering governing equations from data is an important topic at the intersection of AI and science. One branch of methods relies on search algorithms and has achieved interpretable results. Deep Symbolic Regression (DSR) (Petersen et al., 2019) employs a novel risk-seeking policy gradient to train a recurrent neural network, which emits a distribution over tractable mathematical expressions. Mundhenk et al. (2021) utilize neural-guided search to generate starting populations for a random restart genetic programming component, aiming to solve symbolic regression and other symbolic optimization problems. Symbolic Physics Learner (SPL) (Sun et al., 2022) machine leverages a Monte Carlo Tree Search (MCTS) agent to construct optimal expression trees, which interpret mathematical operations and variables. PySR (Cranmer, 2023) adopts a multi-population evolutionary algorithm and a unique evolve-simplify-optimize loop to accelerate the discovery of symbolic models. However, a limitation of such methods is their low computational efficiency when the search space is large.

Another branch of methods leverages deep learning to improve the efficiency of equation discovery. SINDy (Brunton et al., 2016) employs sparse regression to identify equation forms that are both accurate and concise. Building upon SINDy, Champion et al. (2019) further utilize a deep autoencoder network to transform coordinates into a reduced space where the dynamics can be sparsely represented. Weak SINDy (Messenger & Bortz, 2021) replaces pointwise derivative approximations with linear transformations and variance reduction techniques to enhance the robustness of SINDy against noise. NeSymReS (Biggio et al., 2021) pre-trains a Transformer to predict from an unbounded set of equations. These methods still require assumptions about key relevant terms of the equation skeleton and fail to incorporate scientific prior knowledge to narrow the search space for equations.

**Applications of symmetry.** Symmetry plays an important role in both traditional mathematical physics problems and the field of deep learning. We summarize related works in Appendix A.

## 3 PRELIMINARY

Before introducing the method, we will first briefly present some preliminary knowledge concerning partial differential equations and their Lie point symmetries. For more details, please refer to the textbook (Olver, 1993). **For readers unfamiliar with the theory related to Lie groups and Lie algebras, we strongly recommend referring to the concrete examples in Appendix B for an intuitive understanding of these concepts.**

**Partial differential equations.** Let the independent variable $x \in X = \mathbb{R}^p$ and the dependent variable $u \in U = \mathbb{R}^q$. We denote the $k$-th order derivative of $u$ with respect to $x$ as $u_J^\alpha = \frac{\partial^k u^\alpha}{\partial x^{j_1} \partial x^{j_2} \dots \partial x^{j_k}} \in U_k$, where $\alpha \in \{1, \dots, q\}$, $J = (j_1, \dots, j_k)$, and $j_i \in \{1, \dots, p\}$. Furthermore, all derivatives of $u$ with respect to $x$ up to order $n$ are denoted as $u^{(n)} \in U^{(n)} = U \times U_1 \times \dots \times U_n$. Based on the above concepts, we can define a system of $n$-th order partial differential equations as $F(x, u^{(n)}) = 0$, where $F : X \times U^{(n)} \to \mathbb{R}^l$. Its solution is given by a smooth function $f : X \to U$.

**Lie point symmetries.** The solution to the system of partial differential equations $F(x, u^{(n)}) = 0$ can also be represented by the graph $\Gamma_f = \{(x, f(x)) : x \in X\}$ of the function $f : X \to U$. Let the Lie group $G$ act on $X \times U$. We say that $G$ is a symmetry group of $F(x, u^{(n)}) = 0$ if, for any solution $f$ with its graph $\Gamma_f$ and any group element $g \in G$, $g \cdot \Gamma_f = \{(\tilde{x}, \tilde{u}) = g \cdot (x, u) : (x, u) \in \Gamma_f\}$ is the graph $\Gamma_{\tilde{f}}$ of another solution $\tilde{f}$.

The Lie point symmetries of partial differential equations can be restated more simply if we introduce the concept of the prolonged group action, which acts on $X \times U^{(n)}$. Denote the action of a group element $g \in G$ at a point $(x, u) \in X \times U$ as $(\tilde{x}, \tilde{u}) = g \cdot (x, u)$. Then, we define the $n$-th order prolongation of $g$ at the point $(x, u^{(n)}) \in X \times U^{(n)}$ as $\mathrm{pr}^{(n)} g \cdot (x, u^{(n)}) = (\tilde{x}, \tilde{u}^{(n)})$, where $\tilde{u}^{(n)}$ consists of all derivatives of $\tilde{u}$ with respect to $\tilde{x}$ up to order $n$. $G$ is a symmetry group of $F(x, u^{(n)}) = 0$ means that for any solution $u = f(x)$ and any group element $g \in G$, $F(\mathrm{pr}^{(n)} g \cdot (x, u^{(n)})) = 0$ holds.

**Infinitesimal criteria.** Suppose the Lie group $G$ corresponds to the Lie algebra $\mathfrak{g}$, which can be associated via the exponential map $\exp : \mathfrak{g} \rightarrow G$. The infinitesimal group action $\mathbf{v} \in \mathfrak{g}$ at the point $(x, u) \in X \times U$ is defined as $\mathbf{v}|_{(x,u)} = \frac{\mathrm{d}}{\mathrm{d}\epsilon}\big|_{\epsilon=0} [\exp(\epsilon \mathbf{v}) \cdot (x, u)]$. Note that $\mathbf{v}$ is expressed in terms of the partial differential operator $\nabla$ as its special basis, which indicates that it can directly act on functions defined on $X \times U$. Taking the SO(2) group $\epsilon \cdot (x, u) = (x \cos \epsilon - u \sin \epsilon, x \sin \epsilon + u \cos \epsilon)$ as an example, its infinitesimal group action is $\mathbf{v}|_{(x,u)} = -u\frac{\partial}{\partial x} + x\frac{\partial}{\partial u}$.

Similarly, we define the $n$-th order prolongation of $\mathbf{v} \in \mathfrak{g}$ at the point $(x, u^{(n)}) \in X \times U^{(n)}$ as $\mathrm{pr}^{(n)}\mathbf{v}\big|_{(x,u^{(n)})} = \frac{\mathrm{d}}{\mathrm{d}\epsilon}\big|_{\epsilon=0} \big\{ \mathrm{pr}^{(n)} [\exp(\epsilon\mathbf{v})] \cdot (x, u^{(n)}) \big\}$. Then, according to Theorem 2.31 in the textbook (Olver, 1993), $G$ is a symmetry group of $F(x, u^{(n)}) = 0$ if, for every $\mathbf{v} \in \mathfrak{g}$, $\mathrm{pr}^{(n)}\mathbf{v}\big[F(x, u^{(n)})\big] = 0$ whenever $F(x, u^{(n)}) = 0$.

## 4 METHOD

In short, we explore the use of prior knowledge about the symmetry group $G$ to guide the discovery of governing PDEs $F(x, u^{(n)}) = 0$ from the dataset $\mathcal{D} = \{(x[i], u[i])\}_{i=1}^{N}$. In Section 4.1, we prolong the infinitesimal generators of the symmetry group and compute the corresponding differential invariants. In Section 4.2, we discuss integrating differential invariants with existing equation discovery methods and provide a proposition to demonstrate that our approach is both correct and complete. In Section 4.3, we take SINDy (Brunton et al., 2016) as an example to showcase the theoretical advantages of our method over other symmetry-guided equation discovery approaches, such as EquivSINDy-c and EquivSINDy-r (Yang et al., 2024b). Figure 2 provides an intuitive summary of our differential invariant-based equation discovery pipeline. **For readers unfamiliar with the theory of Lie groups and Lie algebras, we strongly recommend referring to the concrete examples in Appendix C to intuitively understand the derivation process of differential invariants.**

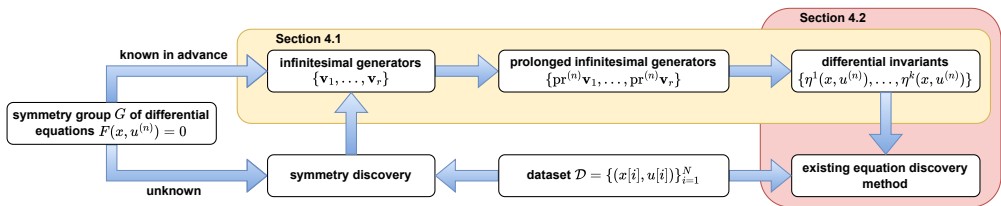

Figure 2: Pipeline of our differential invariant-based equation discovery method.

### 4.1 CALCULATION OF DIFFERENTIAL INVARIANTS

Differential invariants refer to quantities that remain unchanged under the action of a prolonged group. Definition 2.51 in the textbook (Olver, 1993) provides a formal definition of differential invariants, which we briefly restate as follows.

**Definition 4.1** *Let $G$ be a Lie group acting on $X \times U$. An $n$-th order differential invariant of $G$ is a smooth function $\eta : X \times U^{(n)} \rightarrow \mathbb{R}$ such that $\eta$ is an invariant under the prolonged group action $\mathrm{pr}^{(n)}G$:*

$$\forall g \in G, (x, u^{(n)}) \in X \times U^{(n)} : \quad \eta(\mathrm{pr}^{(n)}g \cdot (x, u^{(n)})) = \eta(x, u^{(n)}). \tag{1}$$

We now discuss how to find the differential invariants of a Lie group $G$. This problem can be formalized as follows: given the infinitesimal generators $\{\mathbf{v}_1, \dots, \mathbf{v}_r\}$ of the Lie group $G$, we seek a complete set of functionally independent $n$-th order differential invariants $\{\eta^1(x, u^{(n)}), \dots, \eta^k(x, u^{(n)})\}$ for $\mathrm{pr}^{(n)}G$ (functionally independent: they cannot be expressed as combinations of each other).

The first thing we need to do is derive the $n$-th order prolongation $\{\mathrm{pr}^{(n)}\mathbf{v}_1, \dots, \mathrm{pr}^{(n)}\mathbf{v}_r\}$ of the infinitesimal generators. Consider an infinitesimal group action on $X \times U = \mathbb{R}^p \times \mathbb{R}^q$ in the form:

$$\mathbf{v} = \sum_{i=1}^{p} \xi^i(x, u)\frac{\partial}{\partial x^i} + \sum_{\alpha=1}^{q} \phi_\alpha(x, u)\frac{\partial}{\partial u^\alpha}. \tag{2}$$

Then, according to Theorem 2.36 in the textbook (Olver, 1993), its $n$-th order prolongation is:

$$\mathrm{pr}^{(n)}\mathbf{v} = \mathbf{v} + \sum_{\alpha=1}^{q}\sum_{J}\phi_{\alpha}^{J}(x, u^{(n)})\frac{\partial}{\partial u_J^{\alpha}}, \tag{3}$$

where the coefficients are determined by:

$$\phi_{\alpha}^{J}(x, u^{(n)}) = \mathrm{D}_J\left(\phi_{\alpha} - \sum_{i=1}^{p}\xi^i u_i^{\alpha}\right) + \sum_{i=1}^{p}\xi^i u_{J,i}^{\alpha}. \tag{4}$$

Here, $J = (j_1, \ldots, j_k)$ with $j_i = 1, \ldots, p$ and $k = 1, \ldots, n$, $u_i^{\alpha} = \frac{\partial u^{\alpha}}{\partial x^i}$, and $u_{J,i}^{\alpha} = \frac{\partial u_J^{\alpha}}{\partial x^i} = \frac{\partial^{k+1}u^{\alpha}}{\partial x^i \partial x^{j_1}\ldots\partial x^{j_k}}$. Note that $\mathrm{D}_J$ denotes the total derivative. For a smooth function $P(x, u^{(n)})$, its relationship with partial derivatives is given by $\mathrm{D}_i P = \frac{\partial P}{\partial x^i} + \sum_{\alpha=1}^{q}\sum_{J}u_{J,i}^{\alpha}\frac{\partial P}{\partial u_J^{\alpha}}$. Taking the infinitesimal group action $\mathbf{v} = -u\frac{\partial}{\partial x} + x\frac{\partial}{\partial u}$ of the SO(2) group as an example, its first-order prolongation is $\mathrm{pr}^{(1)}\mathbf{v} = \mathbf{v} + \phi^x(x, u, u_x)\frac{\partial}{\partial u_x}$, where $\phi^x(x, u, u_x) = \mathrm{D}_x(x + uu_x) - uu_{xx} = 1 + u_x^2$.

Next, we derive the $n$-th order differential invariants based on the prolonged infinitesimal generators. According to the infinitesimal criteria introduced in Section 3, Equation (1) is equivalent to:

$$\mathrm{pr}^{(n)}\mathbf{v}\left[\eta(x, u^{(n)})\right] = \sum_{i=1}^{p}\xi^i(x, u)\frac{\partial\eta}{\partial x^i} + \sum_{\alpha=1}^{q}\phi_{\alpha}(x, u)\frac{\partial\eta}{\partial u^{\alpha}} + \sum_{\alpha=1}^{q}\sum_{J}\phi_{\alpha}^{J}(x, u^{(n)})\frac{\partial\eta}{\partial u_J^{\alpha}} = 0. \tag{5}$$

Then, we construct the characteristic equations:

$$\frac{\mathrm{d}x^i}{\xi^i(x, u)} = \frac{\mathrm{d}u^{\alpha}}{\phi_{\alpha}(x, u)} = \frac{\mathrm{d}u_J^{\alpha}}{\phi_{\alpha}^{J}(x, u^{(n)})}, \tag{6}$$

for all $i = 1, \ldots, p$, $\alpha = 1, \ldots, q$, and $J = (j_1, \ldots, j_k)$ with $j_i = 1, \ldots, p$ and $k = 1, \ldots, n$. The integration constants of the general solution to the characteristic equations yield the differential invariants:

$$\eta^1(x, u^{(n)}) = c_1, \ldots, \eta^k(x, u^{(n)}) = c_k. \tag{7}$$

In the case of multiple prolonged infinitesimal generators, we solve the corresponding characteristic equations jointly. Taking the SO(2) group as an example again, the first-order prolongation of its infinitesimal generator is $\mathrm{pr}^{(1)}\mathbf{v} = -u\frac{\partial}{\partial x} + x\frac{\partial}{\partial u} + (1 + u_x^2)\frac{\partial}{\partial u_x}$. We construct the characteristic equation as $\frac{\mathrm{d}x}{-u} = \frac{\mathrm{d}u}{x} = \frac{\mathrm{d}u_x}{1+u_x^2}$. The constants obtained by integration are $\eta^1(x, u, u_x) = \sqrt{x^2 + u^2}$ and $\eta^2(x, u, u_x) = \frac{xu_x - u}{uu_x + x}$, which constitute the first-order differential invariants of the SO(2) group.

## 4.2 Governing Equation Discovery Based on Differential Invariants

Existing equation discovery methods typically follow the paradigm of first specifying the equation skeleton and then optimizing the parameters. When manually specifying the equation skeleton, the challenge lies in selecting the relevant terms. Including too many irrelevant terms leads to excessive computational costs and reduced accuracy, while omitting key terms makes it theoretically impossible for the algorithm to achieve the correct solution. This limitation becomes even more pronounced in partial differential equation discovery, as compared to $X \times U$, $X \times U^{(n)}$ usually constitutes a much larger search space with more candidate terms to choose from.

Our method aims to use symmetry to guide the selection of relevant terms. We hope that this selection approach, while respecting symmetry, can provide a relatively concise search space without losing expressive power. Proposition 2.56 in the textbook (Olver, 1993) provides the inspiration, which we briefly restate as follows.

**Proposition 4.1** *Let $G$ be a Lie group acting on $X \times U$, and $\eta^1(x, u^{(n)}), \ldots, \eta^k(x, u^{(n)})$ be a complete set of functionally independent $n$-th order differential invariants. An $n$-th order differential equation $F(x, u^{(n)}) = 0$ admits $G$ as a symmetry group if and only if there is an equivalent equation*

$$\widetilde{F}(\eta^1(x, u^{(n)}), \ldots, \eta^k(x, u^{(n)})) = 0 \tag{8}$$

*involving only the differential invariants of $G$.*

Therefore, we first use the procedure in Section 4.1 to compute differential invariants based on the symmetry group, which serve as all the relevant terms. Then, we can choose any existing equation discovery method (Brunton et al., 2016; Champion et al., 2019; Messenger & Bortz, 2021; Biggio et al., 2021) to explicitly solve for $\widetilde{F}$. Our approach does not interfere with the core of these methods, except for providing the selection of relevant terms, which means it is plug-and-play. Proposition 4.1 theoretically guarantees that this substitution approach strictly adheres to the symmetry prior while ensuring that the equation skeleton is not missing potential solutions due to the omission of relevant terms. When the symmetry is unknown, we can first employ symmetry discovery methods (Yang et al., 2024a; Ko et al., 2024; Shaw et al., 2024) to obtain infinitesimal generators from the data and then implement the aforementioned equation discovery process.

Note that we do not need to exhaustively provide all infinitesimal generators of the symmetry group. In most cases, we might miss some infinitesimal generators due to reasons such as errors in symmetry detection, but this does not affect the correctness of the equation discovery results. This is because if a Lie group $G$ is the symmetry group of a differential equation, so is any subgroup $\widetilde{G} \subseteq G$. In fact, each additional correct infinitesimal generator we provide reduces the complete set of functionally independent differential invariants, which leads to a smaller and more accurate search space for the governing equation. In Table 1, we use the Lie point symmetries of the KdV, KS, and Burgers equations mentioned by Ko et al. (2024) as examples to demonstrate the complete set of functionally independent differential invariants corresponding to different numbers of infinitesimal generators.

Table 1: The complete set of functionally independent differential invariants corresponding to different numbers of provided infinitesimal generators. For detailed calculation steps, refer to Appendix C.1.

| Provided infinitesimal generators | Complete set of functionally independent differential invariants |
|---|---|
| $\emptyset$ | $\{t, x, u, u_t, u_x, u_{xx}, u_{xxx}, u_{xxxx}\}$ |
| $\{\partial_x\}$ | $\{t, u, u_t, u_x, u_{xx}, u_{xxx}, u_{xxxx}\}$ |
| $\{\partial_x, \partial_t\}$ | $\{u, u_t, u_x, u_{xx}, u_{xxx}, u_{xxxx}\}$ |
| $\{\partial_x, \partial_t, t\partial_x + \partial_u\}$ | $\{u_t + uu_x, u_x, u_{xx}, u_{xxx}, u_{xxxx}\}$ |

### 4.3 Example Algorithm: DI-SINDy

---

**Algorithm 1** DI-SINDy (SINDy based on Differential Invariants)

---

**Input:** Dataset $\mathcal{D} = \{(x[i], u[i])\}_{i=1}^N$, prolongation order $n$, infinitesimal generators of the symmetry group $V(\mathfrak{g}) = \{\mathbf{v}_1, \ldots, \mathbf{v}_r\}$.
**Output:** Explicit governing equation $F(x, u^{(n)}) = 0$.
**Execute:**
Estimate the derivatives of $u$ with respect to $x$ using the central difference method, resulting in the prolonged dataset $\mathrm{pr}^{(n)}\mathcal{D} = \{(x[i], u^{(n)}[i])\}_{i=1}^N$.
**if** $V(\mathfrak{g}) = \emptyset$ **then**
  Use the method of symmetry discovery to obtain the infinitesimal generators $V(\mathfrak{g}) = \{\mathbf{v}_1, \ldots, \mathbf{v}_r\}$ of the symmetry group from $\mathrm{pr}^{(n)}\mathcal{D}$.
**end if**
Derive the prolonged infinitesimal generators $\{\mathrm{pr}^{(n)}\mathbf{v}_1, \ldots, \mathrm{pr}^{(n)}\mathbf{v}_r\}$ according to Equations (2) to (4).
Compute differential invariants $\{\eta^1(x, u^{(n)}), \ldots, \eta^k(x, u^{(n)})\}$ according to Equations (5) to (7).
For the equation skeleton $\eta^k(x, u^{(n)}) = W\Theta(\eta^1(x, u^{(n)}), \ldots, \eta^{k-1}(x, u^{(n)}))$, optimize the coefficient matrix $W$ using SINDy based on $\mathrm{pr}^{(n)}\mathcal{D}$.
**Return** $F(x, u^{(n)}) = \eta^k(x, u^{(n)}) - W\Theta(\eta^1(x, u^{(n)}), \ldots, \eta^{k-1}(x, u^{(n)})) = 0$.

---

Now our method can be summarized as follows. First, we use symmetry discovery methods to obtain infinitesimal generators from the dataset if the symmetries are not known a priori. Then, we derive the prolonged infinitesimal generators and compute the differential invariants based on them. Finally, we select the relevant terms of the equation skeleton from the differential invariants and employ existing equation discovery methods to obtain the explicit governing equation. Taking

SINDy (Brunton et al., 2016) based on Differential Invariants (DI-SINDy) as an example, we outline the overall workflow in Algorithm 1.

The EquivSINDy-c and EquivSINDy-r methods proposed by Yang et al. (2024b) also attempt to use symmetry to guide SINDy in discovering governing equations of the form $h(x) = W\Theta(x)$. However, for EquivSINDy-c, it cannot handle nonlinear cases, and Proposition 4.2 in the original paper (Yang et al., 2024b) specifies that $\Theta(x)$ can only be chosen as polynomials. Additionally, the constrained parameter space of $W$ reduces the expressive power of the equation skeleton. On the other hand, the necessity and sufficiency of Proposition 4.1 in this paper guarantee that DI-SINDy's skeleton can fully express all equations satisfying the symmetry, and $\Theta(x)$ can be freely selected, thereby addressing the limitations of EquivSINDy-c. Compared to EquivSINDy-r, which incorporates symmetry loss as a regularization term into SINDy's loss function, DI-SINDy ensures that the equation skeleton strictly adheres to symmetry without requiring hyperparameter tuning for regularization coefficients. Overall, DI-SINDy holds significant theoretical advantages over related works, thanks to its intrinsic ability to "losslessly" compress the equation search space based on symmetry.

## 5 EXPERIMENT

### 5.1 EXPERIMENTAL SETUP

We evaluate our method using the Korteweg-de Vries (KdV) equation, the Kuramoto-Shivashinsky (KS) equation, the Burgers equation, and the nKdV equation from Ko et al. (2024). In Table 2, we present their explicit equations, the infinitesimal generators of their symmetry groups, and the corresponding differential invariants (detailed calculation steps are provided in Appendix C), where the prolongation order is specified as fourth-order. We assume the symmetries are known a priori, and the experimental task is to automatically discover the governing equations from the generated data. The infinitesimal generators provided here are all sufficiently simple to be easily obtained by existing symmetry discovery methods. We provide the data generation process in Appendix D.

Table 2: Explicit expressions, infinitesimal generators of symmetry groups, and corresponding differential invariants for the KdV, KS, Burgers, and nKdV equations Ko et al. (2024).

| Name | Equation | Infinitesimal generators | Differential invariants |
|------|----------|--------------------------|-------------------------|
| KdV | $u_t + uu_x + u_{xxx} = 0$ | | |
| KS | $u_t + u_{xx} + u_{xxxx} + uu_x = 0$ | $\{\frac{\partial}{\partial x}, \frac{\partial}{\partial t}, t\frac{\partial}{\partial x} + \frac{\partial}{\partial u}\}$ | $\{u_t + uu_x, u_x, u_{xx}, u_{xxx}, u_{xxxx}\}$ |
| Burgers | $u_t + uu_x - \nu u_{xx} = 0$ | | |
| nKdV | $e^{-\frac{t}{t_0}}u_t + uu_x + u_{xxx} = 0$ | $\{\frac{\partial}{\partial x}, e^{-\frac{t}{t_0}}\frac{\partial}{\partial t}, t_0(e^{\frac{t}{t_0}} - 1)\frac{\partial}{\partial x} + \frac{\partial}{\partial u}\}$ | $\{e^{-\frac{t}{t_0}}u_t + uu_x, u_x, u_{xx}, u_{xxx}, u_{xxxx}\}$ |

Taking DI-SINDy presented in Algorithm 1 as an example, we compare it with SINDy (Brunton et al., 2016) and EquivSINDy-r (Yang et al., 2024b). The Lie point symmetry of PDEs is typically nonlinear, which renders EquivSINDy-c inapplicable—hence we exclude it from the comparison. The idea behind EquivSINDy-r is to incorporate the infinitesimal criterion of the symmetry group as a regularization term into the objective function of SINDy, thereby softly constraining the equation skeleton to adhere to the symmetry. The original paper (Yang et al., 2024b) only provides the form of the regularization term for ODE cases. To extend it to PDE scenarios for comparison, we adopt the infinitesimal criterion of Lie point symmetry introduced in Section 3 as the regularization term:

$$\mathcal{L}_{symm} = \mathbb{E}_{x,u}\left\{\sum_{\mathbf{v} \in V(\mathfrak{g})}\left\|\mathrm{pr}^{(n)}\mathbf{v}\left[F(x, u^{(n)})\right]\right\|^2\right\}, \tag{9}$$

where $V(\mathfrak{g})$ is the set of infinitesimal generators of the symmetry group, and $F$ represents the equation skeleton of SINDy. Then, the overall objective function of EquivSINDy-r is:

$$\mathcal{L}_{total} = \mathcal{L}_{SINDy} + \lambda \cdot \mathcal{L}_{symm}. \tag{10}$$

For a comprehensive comparison, we will traverse the regularization weight hyperparameter $\lambda = \{10^{-3}, 10^{-2}, 10^{-1}\}$.

As described in Algorithm 1, the relevant terms of DI-SINDy are selected as the set of differential invariants shown in Table 2, and the function library $\Theta$ is specified as linear terms. For SINDy and EquivSINDy-r, we define the equation skeleton of the KdV, KS, and Burgers equations as $u_t = W\Theta(u, u_x, u_{xx}, u_{xxx}, u_{xxxx})$, and the equation skeleton of the nKdV equation as $e^{-\frac{t}{t_0}} u_t = W\Theta(u, u_x, u_{xx}, u_{xxx}, u_{xxxx})$, where $\Theta$ contains terms up to second order. It can be observed that the baseline methods require strong prior assumptions about the equation skeleton during the experimental preparation phase, even though we have manually specified relatively simple forms for them that include the ground truth. More implementation details can be found in Appendix E.

## 5.2 QUANTITATIVE METRICS AND RESULT ANALYSIS

After training with SINDy and its variant methods, we get explicit equations such as $u_t = W\Theta(u, u_x, \dots)$ (for KdV, KS, and Burgers equations) or $e^{-\frac{t}{t_0}} u_t = W\Theta(u, u_x, \dots)$ (for the nKdV equation). In practice, the coefficient matrix is obtained via element-wise multiplication $W = C \odot M$, where $C$ represents the values of each term's coefficient, and the binary mask matrix $M$ indicates whether each term is retained (1 for retained, 0 for discarded). We follow the quantitative metrics introduced by Yang et al. (2024b), which we restate as follows. We consider the discovery of an equation successful if the retained terms in the final result are correct and complete (formally, $M = M^*$, where $M^*$ is the ground truth of the binary mask matrix). We run each experiment 50 times and calculate its **success rate**, which is the most important quantitative metric for explicit equation discovery, as it reflects whether the model can correctly identify the interaction relationships between variables. Furthermore, we use the RMSE of the coefficient matrix, $\sqrt{\frac{1}{n} \sum_{i=1}^{n} \|W - W^*\|^2}$, to evaluate the accuracy of equation discovery, where $n$ is the number of runs, and $W^*$ is the ground truth of the coefficient matrix. We report **RMSE (successful)** and **RMSE (all)**, which represent the RMSE for successful runs and all runs, respectively.

Table 3: Success rates and RMSE of different SINDy-based methods for the KdV, KS, Burgers, and nKdV equations. All experimental results are averaged over 50 runs. RMSE is presented in the format of mean ± std.

| Name | Method | Success rate (↑) | RMSE (successful) (↓) | RMSE (all) (↓) |
|------|--------|------------------|------------------------|-----------------|
| KdV | SINDy | 72% | $(2.24 \pm 0.51) \times 10^{-1}$ | $(4.42 \pm 3.51) \times 10^{-1}$ |
| | EquivSINDy-r ($\lambda = 10^{-3}$) | 72% | $(2.23 \pm 0.51) \times 10^{-1}$ | $(4.41 \pm 3.51) \times 10^{-1}$ |
| | EquivSINDy-r ($\lambda = 10^{-2}$) | 74% | $(2.18 \pm 0.50) \times 10^{-1}$ | $(9.28 \pm 14.01) \times 10^{-2}$ |
| | EquivSINDy-r ($\lambda = 10^{-1}$) | 82% | $(1.66 \pm 0.37) \times 10^{-1}$ | $(3.16 \pm 3.22) \times 10^{-1}$ |
| | DI-SINDy (Ours) | **100**% | $(\mathbf{2.71 \pm 2.44}) \times 10^{-2}$ | $(\mathbf{2.71 \pm 2.44}) \times 10^{-2}$ |
| KS | SINDy | 0% | N/A | $1.00 \pm 0.00$ |
| | EquivSINDy-r ($\lambda = 10^{-3}$) | 0% | N/A | $1.00 \pm 0.00$ |
| | EquivSINDy-r ($\lambda = 10^{-2}$) | 0% | N/A | $1.00 \pm 0.00$ |
| | EquivSINDy-r ($\lambda = 10^{-1}$) | 0% | N/A | $1.00 \pm 0.00$ |
| | DI-SINDy (Ours) | **100**% | $(\mathbf{6.18 \pm 0.37}) \times 10^{-2}$ | $(\mathbf{6.18 \pm 0.37}) \times 10^{-2}$ |
| Burgers | SINDy | 4% | $(2.11 \pm 0.14) \times 10^{-2}$ | $(1.52 \pm 2.34) \times 10^{-1}$ |
| | EquivSINDy-r ($\lambda = 10^{-3}$) | 16% | $(2.59 \pm 0.42) \times 10^{-2}$ | $(1.86 \pm 4.12) \times 10^{-1}$ |
| | EquivSINDy-r ($\lambda = 10^{-2}$) | 68% | $(8.06 \pm 3.38) \times 10^{-3}$ | $(9.78 \pm 38.08) \times 10^{-2}$ |
| | EquivSINDy-r ($\lambda = 10^{-1}$) | 78% | $(9.68 \pm 3.89) \times 10^{-4}$ | $(7.03 \pm 35.62) \times 10^{-2}$ |
| | DI-SINDy (Ours) | **98**% | $(\mathbf{2.66 \pm 1.32}) \times 10^{-4}$ | $(\mathbf{4.02 \pm 9.62}) \times 10^{-4}$ |
| nKdV | SINDy | 20% | $(3.77 \pm 0.14) \times 10^{-1}$ | $(8.75 \pm 2.49) \times 10^{-1}$ |
| | EquivSINDy-r ($\lambda = 10^{-3}$) | 20% | $(3.76 \pm 0.14) \times 10^{-1}$ | $(8.75 \pm 2.50) \times 10^{-1}$ |
| | EquivSINDy-r ($\lambda = 10^{-2}$) | 22% | $(3.62 \pm 0.13) \times 10^{-1}$ | $(8.60 \pm 2.64) \times 10^{-1}$ |
| | EquivSINDy-r ($\lambda = 10^{-1}$) | 44% | $(2.70 \pm 0.19) \times 10^{-1}$ | $(6.79 \pm 3.63) \times 10^{-1}$ |
| | DI-SINDy (Ours) | **100**% | $(\mathbf{5.05 \pm 3.84}) \times 10^{-2}$ | $(\mathbf{5.05 \pm 3.84}) \times 10^{-2}$ |

The success rates and RMSE of different SINDy-based methods are presented in Table 3. For the KdV, Burgers, and nKdV equations, EquivSINDy-r, with its soft symmetry constraints, significantly improves both the success rate and accuracy compared to SINDy, while our DI-SINDy further increases the success rate to nearly 100%. Notably, both SINDy and EquivSINDy-r fail for the KS equation, as the KS equation involves a fourth-order derivative term, making finite difference methods prone to large errors in the presence of noise. In contrast, DI-SINDy, benefiting from a smaller

search space, can still accurately identify the correct equation form, demonstrating stronger robustness.

Beyond quantitative advantages, as discussed in Section 5.1, DI-SINDy employs differential invariants as candidate terms, unlike SINDy and EquivSINDy-r, which rely on manually specified equation skeletons (e.g., for the nKdV equation, the term $e^{-\frac{t}{t_0}} u_t$ is difficult to guess, whereas differential invariants naturally guide its inclusion). Additionally, the performance of EquivSINDy-r is sensitive to the regularization weight $\lambda$, while DI-SINDy eliminates the need for hyperparameter tuning.

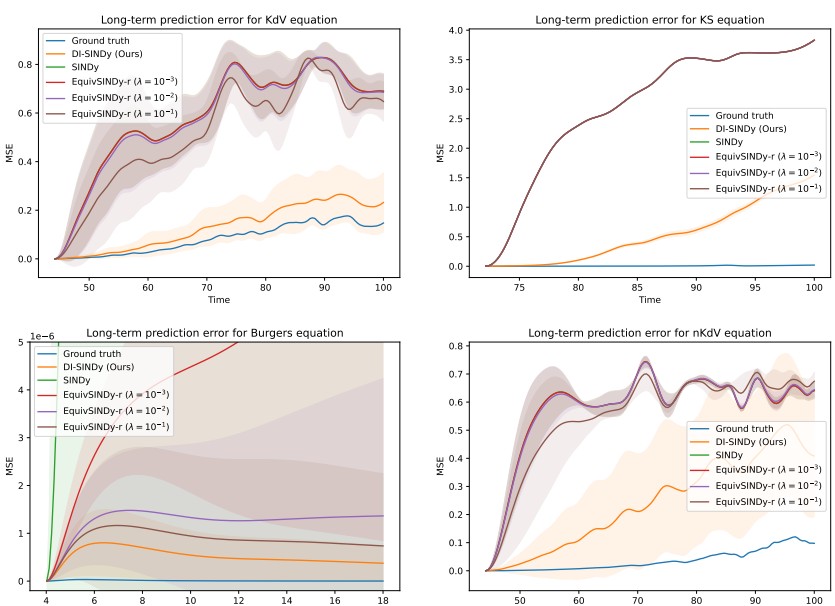

Figure 3: Long-term prediction errors of different SINDy-based methods for the KdV, KS, Burgers, and nKdV equations. The MSE at each time step is averaged over $4$ initial conditions and $50$ runs, with the shaded area representing the standard deviation.

We further numerically integrate the discovered explicit equations for the $4$ initial conditions in the test dataset and calculate the MSE against their corresponding true trajectories, which we refer to as the **long-term prediction error**. In Figure 3, we visualize the long-term prediction errors of all SINDy-based methods for the KdV, KS, Burgers, and nKdV equations as a function of the integration time steps. We use the ground-truth equation form as the benchmark (blue lines), for which the long-term prediction error primarily stems from finite differences and numerical integration. For the KdV and nKdV equations, the error curves of SINDy and EquivSINDy-r ($\lambda = 10^{-3}$) almost overlap, while for the KS equation, the error curves of SINDy and EquivSINDy-r with all $\lambda$ values nearly coincide. This is due to the minimal differences in their discovered explicit equations, which can be verified by the numerical results in Table 3. For all PDEs, our DI-SINDy achieves significantly lower long-term prediction errors than baselines, further validating the accuracy of its equation discovery results. Additional experimental results, including robustness analysis on datasets and symmetric noise, as well as solutions for high-dimensional dynamic systems, are provided in Appendix F.

## 5.3 ROBUST ANALYSIS AGAINST DATASET AND INFINITESIMAL GENERATOR NOISE

We test the stability of our DI-SINDy method through robustness experiments under conditions such as dataset noise and inaccurate symmetry priors. First, we progressively increase the noise level ($\sigma = \{3 \times 10^{-3}, 5 \times 10^{-3}\}$) on the Burgers equation dataset. The results show that DI-SINDy maintains a high identification success rate and prediction accuracy even when the standard SINDy method fails completely, demonstrating its strong robustness against real-world data interference. Second, we simulate scenarios with deviations in the infinitesimal generator (relative perturbation amplitude $\delta = \{10^{-1}, 3 \times 10^{-1}, 5 \times 10^{-1}\}$) and find that DI-SINDy also exhibits high tolerance,

outperforming symmetry-agnostic SINDy when $\delta \leq 3 \times 10^{-1}$, highlighting its potential for integration with symmetry discovery methods. Detailed experimental setups and result analyses are provided in Appendices F.1 and F.2.

### 5.4 HIGH-DIMENSIONAL CASE: GOVERNING EQUATION DISCOVERY FOR REACTION-DIFFUSION SYSTEM

In the high-dimensional case study of reaction-diffusion systems, we validate the capability of the DI-SINDy method in discovering complex partial differential equations. By leveraging differential invariant theory, the method compresses the original search space from hundreds of terms to just 9 terms composed of 5 key invariants, constructing a concise equation skeleton. Experimental results demonstrate that DI-SINDy accurately uncovers the governing equations of the system, including the correct structures of nonlinear reaction and diffusion terms, proving its effectiveness in handling high-dimensional dynamical systems. In contrast, traditional SINDy and EquivSINDy-r methods fail due to the curse of dimensionality. Detailed experimental setups, theoretical derivations, and result analyses are provided in Appendix F.3.

### 5.5 DIFFERENTIAL INVARIANTS GUIDE TRANSFORMER-BASED GOVERNING EQUATION DISCOVERY

Furthermore, we validate the universality of combining differential invariants with the Transformer-based symbolic regression method (E2E) (Kamienny et al., 2022) and propose the DI-E2E method. Experiments demonstrate that, compared to the original E2E method, our DI-E2E achieves significant performance improvements across the KdV, KS, Burgers, nKdV, and reaction-diffusion equations. It not only exhibits higher prediction accuracy and better stability but also requires shorter inference time. Particularly in complex high-dimensional reaction-diffusion systems, our DI-E2E achieves an almost perfect fit ($R^2 \approx 1$), whereas the original E2E method fails completely. Detailed experimental setups and result analyses are provided in Appendix F.4.

## 6 CONCLUSION

Overall, our method addresses several pain points in existing equation discovery approaches. For the large search space of PDEs, most methods struggle to identify the correct relevant terms, whereas we overcome this limitation by employing differential invariants. The necessity and sufficiency of Proposition 4.1 show that our method neither loses expressiveness like symmetry-constrained approaches such as EquivSINDy-c, nor violates symmetry principles like regularization-based methods such as EquivSINDy-r. We hope that our method, in conjunction with established symmetry discovery efforts, will form a systematic solution for complex scientific problems in the future.

## ETHICS STATEMENT

We have read and adhere to the ICLR Code of Ethics for this work. After careful review, we determine that this paper does not raise any ethical issues. Our research is theoretical in nature and does not involve any human subjects, data collection, or experimental protocols requiring IRB approval. The work is based on publicly available benchmark datasets and does not present any known risks of malicious use, bias, or other societal harms. We have no conflicts of interest to disclose.

## REPRODUCIBILITY STATEMENT

We have made substantial efforts to ensure the reproducibility of our work. The example algorithm is presented in pseudocode in Section 4.3 (Algorithm 1). Detailed experimental settings are provided in Section 5.1. The data generation process is thoroughly described in Appendix D, and additional implementation details are included in Appendix E. Code and data will be released upon acceptance.

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

## A  APPLICATIONS OF SYMMETRY

Symmetry plays an important role in both traditional mathematical physics problems and the field of deep learning. For the mathematical solution of differential equations, symmetry can guide variable substitutions to reduce their order (Olver, 1993; McLachlan, 1995; Ibragimov, 1999; Hydon, 2000; Bluman & Anco, 2008; Bluman, 2010). In recent years, equivariant networks have incorporated symmetry into network architectures, significantly improving performance and generalization in specific scientific and computer vision tasks (Zaheer et al., 2017; Weiler et al., 2018b;a; Kondor & Trivedi, 2018; Wang et al., 2020; Finzi et al., 2021; Satorras et al., 2021; Ruhe et al., 2023; Li et al., 2024; 2025; Hu et al., 2025c). Additionally, symmetry has been introduced into Physics-Informed Neural Networks (PINNs) or used to guide data augmentation to enhance the accuracy of neural PDE solvers (Arora et al., 2024; Lagrave & Tron, 2022; Shumaylov et al., 2024; Li et al., 2022; Zhang et al., 2023; Wang et al., 2025; Akhound-Sadegh et al., 2023; Brandstetter et al., 2022a). Notably, our goal is to discover explicit equations rather than using PINNs to learn the evolution process of PDEs, which means the problem we focus on differs from that of neural PDE solvers. We note that Yang et al. (2025) have also recently attempted to discover equations based on differential invariants, but a direct comparison is exempted due to the concurrent timelines.

## B  EXAMPLE

We take the KdV equation $u_t + uu_x + u_{xxx} = 0$ as an example to intuitively understand the concepts introduced in Section 3. In this case, the independent variables are $(x, t) \in X = \mathbb{R}^2$, and the dependent variable is $u \in U = \mathbb{R}$. Consider the group $G$ acting on $X \times U$, which includes three types of group actions:

$$\begin{cases} \epsilon_1 \cdot (x, t, u) = (x + \epsilon_1, t, u), \\ \epsilon_2 \cdot (x, t, u) = (x, t + \epsilon_2, u), \\ \epsilon_3 \cdot (x, t, u) = (x + \epsilon_3 t, t, u + \epsilon_3). \end{cases} \quad (11)$$

According to the definition $\mathbf{v}|_{(x,u)} = \frac{\mathrm{d}}{\mathrm{d}\epsilon}\big|_{\epsilon=0} [\exp(\epsilon\mathbf{v}) \cdot (x, u)]$, the infinitesimal generators are:

$$\begin{cases} \mathbf{v}_1 = \frac{\partial}{\partial x}, \\ \mathbf{v}_2 = \frac{\partial}{\partial t}, \\ \mathbf{v}_3 = t\frac{\partial}{\partial x} + \frac{\partial}{\partial u}. \end{cases} \quad (12)$$

Assuming $u = f(x, t)$ is a solution to the KdV equation, then under the aforementioned three types of group actions, the graph $\Gamma_f = \{(x, t, f(x, t)) : (x, t) \in X\}$ is transformed into the graphs of the following three functions, respectively:

$$\begin{cases} u^{(1)} = f(x - \epsilon_1, t), \\ u^{(2)} = f(x, t - \epsilon_2), \\ u^{(3)} = f(x - \epsilon_3 t, t) + \epsilon_3. \end{cases} \quad (13)$$

It is easy to verify that if $u = f(x, t)$ satisfies the KdV equation, then $u^{(1)}, u^{(2)}, u^{(3)}$ are also solutions of the equation. Therefore, we call $G$ the symmetry group of the KdV equation.

Note that $u_t^{(3)} = -\epsilon_3 f_x(x - \epsilon_3 t, t) + f_t(x - \epsilon_3 t, t)$. The forms of the other transformed derivatives remain unchanged. Then, we can provide the prolongation of group actions:

$$\begin{cases} \mathrm{pr}^{(n)}\epsilon_1 \cdot (x, t, u, u_t, u_x, \dots) = (x + \epsilon_1, t, u, u_t, u_x, \dots), \\ \mathrm{pr}^{(n)}\epsilon_2 \cdot (x, t, u, u_t, u_x, \dots) = (x, t + \epsilon_2, u, u_t, u_x, \dots), \\ \mathrm{pr}^{(n)}\epsilon_3 \cdot (x, t, u, u_t, u_x, \dots) = (x + \epsilon_3 t, t, u + \epsilon_3, -\epsilon_3 u_x + u_t, u_x, \dots). \end{cases} \quad (14)$$

According to the definition $\mathrm{pr}^{(n)}\mathbf{v}|_{(x,u^{(n)})} = \frac{\mathrm{d}}{\mathrm{d}\epsilon}\big|_{\epsilon=0} \{\mathrm{pr}^{(n)} [\exp(\epsilon\mathbf{v})] \cdot (x, u^{(n)})\}$, the prolongation of the infinitesimal generators are:

$$\begin{cases} \mathrm{pr}^{(n)}\mathbf{v}_1 = \frac{\partial}{\partial x}, \\ \mathrm{pr}^{(n)}\mathbf{v}_2 = \frac{\partial}{\partial t}, \\ \mathrm{pr}^{(n)}\mathbf{v}_3 = t\frac{\partial}{\partial x} + \frac{\partial}{\partial u} - u_x\frac{\partial}{\partial u_t}. \end{cases} \quad (15)$$

Then, we can observe that the infinitesimal criteria $\mathrm{pr}^{(n)}\mathbf{v}_i(u_t + uu_x + u_{xxx}) = 0$ hold for $i = 1, 2, 3$.

## C    DETAILED CALCULATION STEPS OF DIFFERENTIAL INVARIANTS

Consider the case where $X \times U = \mathbb{R}^2 \times \mathbb{R}$, with $(x, t) \in X$ as the independent variables and $u \in U$ as the dependent variable. We specify the highest prolongation order as $n = 4$, so the initial search space consists of the terms $\{t, x, u, u_t, u_x, u_{xx}, u_{xxx}, u_{xxxx}\}$ (for simplicity, we assume the dynamical system is first-order, meaning the highest-order partial derivative of $u$ with respect to $t$ is first-order).

### C.1    KDV, KS, AND BURGERS EQUATIONS

As shown in Table 2, the infinitesimal generators of the symmetry groups for the KdV, KS, and Burgers equations are:

$$\mathbf{v}_1 = \frac{\partial}{\partial x}, \quad \mathbf{v}_2 = \frac{\partial}{\partial t}, \quad \mathbf{v}_3 = t\frac{\partial}{\partial x} + \frac{\partial}{\partial u}. \tag{16}$$

We first compute their fourth-order prolongations. For $\mathrm{pr}^{(4)}\mathbf{v}_1$, we calculate its coefficients from Equation (4):

$$\begin{cases} \phi^t = D_t(-u_x) + u_{tx} = 0, \\ \phi^x = D_x(-u_x) + u_{xx} = 0, \\ \phi^{xx} = D_{xx}(-u_x) + u_{xxx} = 0, \\ \phi^{xxx} = D_{xxx}(-u_x) + u_{xxxx} = 0, \\ \phi^{xxxx} = D_{xxxx}(-u_x) + u_{xxxxx} = 0. \end{cases} \tag{17}$$

Therefore, we have:

$$\mathrm{pr}^{(4)}\mathbf{v}_1 = \mathbf{v}_1 = \frac{\partial}{\partial x}. \tag{18}$$

Similarly, it can be obtained that:

$$\mathrm{pr}^{(4)}\mathbf{v}_2 = \mathbf{v}_2 = \frac{\partial}{\partial t}. \tag{19}$$

The coefficients of $\mathrm{pr}^{(4)}\mathbf{v}_3$ are calculated as follows:

$$\begin{cases} \phi^t = D_t(1 - tu_x) + tu_{tx} = -u_x, \\ \phi^x = D_x(1 - tu_x) + tu_{xx} = 0, \\ \phi^{xx} = D_{xx}(1 - tu_x) + tu_{xxx} = 0, \\ \phi^{xxx} = D_{xxx}(1 - tu_x) + tu_{xxxx} = 0, \\ \phi^{xxxx} = D_{xxxx}(1 - tu_x) + tu_{xxxxx} = 0. \end{cases} \tag{20}$$

This means:

$$\mathrm{pr}^{(4)}\mathbf{v}_3 = \mathbf{v}_3 - u_x\frac{\partial}{\partial u_t} = t\frac{\partial}{\partial x} + \frac{\partial}{\partial u} - u_x\frac{\partial}{\partial u_t}. \tag{21}$$

Substitute $\mathrm{pr}^{(4)}\mathbf{v}_1$ and $\mathrm{pr}^{(4)}\mathbf{v}_2$ into Equation (5):

$$\frac{\partial \eta}{\partial x} = \frac{\partial \eta}{\partial t} = 0. \tag{22}$$

Therefore, the differential invariants do not contain the terms $x$ and $t$. The search space can be narrowed down to $\{u, u_t, u_x, u_{xx}, u_{xxx}, u_{xxxx}\}$.

For $\mathrm{pr}^{(4)}\mathbf{v}_3$, we can construct the characteristic equation as shown in Equation (6) (Note that the term $x$ has already been excluded, so the $t\frac{\partial}{\partial x}$ in $\mathrm{pr}^{(4)}\mathbf{v}_3$ can be ignored):

$$\mathrm{d}u = -\frac{\mathrm{d}u_t}{u_x}. \tag{23}$$

By integrating it, we get:

$$u_t + uu_x = c. \tag{24}$$

Replacing $u$ and $u_t$ in the search space with the integration constant $u_t + uu_x$, we obtain the final differential invariants $\{u_t + uu_x, u_x, u_{xx}, u_{xxx}, u_{xxxx}\}$.

## C.2 NKDV EQUATION

The infinitesimal generators of the symmetry group for the nKdV equation are shown in Table 2 as:

$$\mathbf{v}_1 = \frac{\partial}{\partial x}, \quad \mathbf{v}_2 = e^{-\frac{t}{t_0}}\frac{\partial}{\partial t}, \quad \mathbf{v}_3 = t_0(e^{\frac{t}{t_0}} - 1)\frac{\partial}{\partial x} + \frac{\partial}{\partial u}. \tag{25}$$

The form of $\mathrm{pr}^{(4)}\mathbf{v}_1$ is shown in Equation (18). For $\mathrm{pr}^{(4)}\mathbf{v}_2$, we calculate its coefficients based on Equation (4):

$$\begin{cases} \phi^t = \mathrm{D}_t(-e^{-\frac{t}{t_0}}u_t) + e^{-\frac{t}{t_0}}u_{tt} = \frac{u_t}{t_0}e^{-\frac{t}{t_0}}, \\ \phi^x = \mathrm{D}_x(-e^{-\frac{t}{t_0}}u_t) + e^{-\frac{t}{t_0}}u_{tx} = 0, \\ \phi^{xx} = \mathrm{D}_{xx}(-e^{-\frac{t}{t_0}}u_t) + e^{-\frac{t}{t_0}}u_{txx} = 0, \\ \phi^{xxx} = \mathrm{D}_{xxx}(-e^{-\frac{t}{t_0}}u_t) + e^{-\frac{t}{t_0}}u_{txxx} = 0, \\ \phi^{xxxx} = \mathrm{D}_{xxxx}(-e^{-\frac{t}{t_0}}u_t) + e^{-\frac{t}{t_0}}u_{txxxx} = 0. \end{cases} \tag{26}$$

Then, we have:

$$\mathrm{pr}^{(4)}\mathbf{v}_2 = \mathbf{v}_2 + \frac{u_t}{t_0}e^{-\frac{t}{t_0}}\frac{\partial}{\partial u_t} = e^{-\frac{t}{t_0}}\frac{\partial}{\partial t} + \frac{u_t}{t_0}e^{-\frac{t}{t_0}}\frac{\partial}{\partial u_t}. \tag{27}$$

For $\mathrm{pr}^{(4)}\mathbf{v}_3$, its coefficients are:

$$\begin{cases} \phi^t = \mathrm{D}_t[1 - t_0(e^{\frac{t}{t_0}} - 1)u_x] + t_0(e^{\frac{t}{t_0}} - 1)u_{tx} = -u_x e^{\frac{t}{t_0}}, \\ \phi^x = \mathrm{D}_x[1 - t_0(e^{\frac{t}{t_0}} - 1)u_x] + t_0(e^{\frac{t}{t_0}} - 1)u_{xx} = 0, \\ \phi^{xx} = \mathrm{D}_{xx}[1 - t_0(e^{\frac{t}{t_0}} - 1)u_x] + t_0(e^{\frac{t}{t_0}} - 1)u_{xxx} = 0, \\ \phi^{xxx} = \mathrm{D}_{xxx}[1 - t_0(e^{\frac{t}{t_0}} - 1)u_x] + t_0(e^{\frac{t}{t_0}} - 1)u_{xxxx} = 0, \\ \phi^{xxxx} = \mathrm{D}_{xxxx}[1 - t_0(e^{\frac{t}{t_0}} - 1)u_x] + t_0(e^{\frac{t}{t_0}} - 1)u_{xxxxx} = 0. \end{cases} \tag{28}$$

Then, we get:

$$\mathrm{pr}^{(4)}\mathbf{v}_3 = \mathbf{v}_3 - u_x e^{\frac{t}{t_0}}\frac{\partial}{\partial u_t} = t_0(e^{\frac{t}{t_0}} - 1)\frac{\partial}{\partial x} + \frac{\partial}{\partial u} - u_x e^{\frac{t}{t_0}}\frac{\partial}{\partial u_t}. \tag{29}$$

Similarly to Equation (22), we exclude the variable $x$ based on $\mathrm{pr}^{(4)}\mathbf{v}_1$ and update the search space as $\{t, u, u_t, u_x, u_{xx}, u_{xxx}, u_{xxxx}\}$.

Construct the characteristic equation as shown in Equation (6) based on $\mathrm{pr}^{(4)}\mathbf{v}_2$:

$$e^{\frac{t}{t_0}}\mathrm{d}t = \frac{t_0}{u_t}e^{\frac{t}{t_0}}\mathrm{d}u_t. \tag{30}$$

Integrating it yields the general solution:

$$e^{-\frac{t}{t_0}}u_t = c. \tag{31}$$

By replacing the terms $t$ and $u_t$ with integral constants, we update the search space as $\{e^{-\frac{t}{t_0}}u_t, u, u_x, u_{xx}, u_{xxx}, u_{xxxx}\}$.

For $\mathrm{pr}^{(4)}\mathbf{v}_3$, we construct the characteristic equation as:

$$\mathrm{d}u = -\frac{1}{u_x}e^{-\frac{t}{t_0}}\mathrm{d}u_t. \tag{32}$$

Integral yields:

$$e^{-\frac{t}{t_0}}u_t + uu_x = c. \tag{33}$$

Introducing it into the search space to replace $e^{-\frac{t}{t_0}}u_t$ and $u$, we obtain the final differential invariants as $\{e^{-\frac{t}{t_0}}u_t + uu_x, u_x, u_{xx}, u_{xxx}, u_{xxxx}\}$.

# D  DATA GENERATION

For the generation of the dataset, we follow the setup of Ko et al. (2024), which we restate as follows. In the Burgers equation, $\nu = 0.01$, and in the nKdV equation, $t_0 = 50$. For a one-dimensional PDE $u_t = f(x, t, u, u_x, u_{xx}, \dots)$ defined on $x \in [0, L]$, the initial condition $u(x, t = 0)$ is generated by randomly sampling the coefficients $A_p, l_p, \phi_p$ of the Fourier series $u(x, t = 0) = \sum_{p=1}^{P} A_p \sin(2\pi l_p x / L + \phi_p)$. Then, the spatial derivatives of $u$ with respect to $x$ are estimated using the pseudospectral method, and the temporal evolution $u_t$ is computed from the explicit expression of the equation. After numerically integrating the PDE over $t \in [0, T]$ using an ODE solver, we obtain $N_x \times N_t$ discrete grid points on $[0, L] \times [0, T]$, where $N_x = 256$ and $N_t = 140$. Specifically, the dataset $u(x, t)$ for the Burgers equation is solved indirectly via the heat equation $\phi_t = \phi_{xx}$, which are related through the Cole-Hopf transformation $u = 2\nu \frac{\partial}{\partial x} \ln \phi$. We add multiplicative noise $u' = u \cdot (1 + \epsilon)$ to the vector field $u$ to simulate real-world perturbations, where $\epsilon \sim \mathcal{N}(0, \sigma^2)$. We set the noise level $\sigma = 10^{-2}$ for the KdV and nKdV equations, $\sigma = 10^{-3}$ for Burgers equation, and $\sigma = 10^{-4}$ for the KS equation.

# E  IMPLEMENTATION DETAIL

We select trajectory samples generated from 4 initial conditions in the training dataset for equation discovery and use the L-BFGS optimizer with a learning rate of 0.1 for training. During sparse regression, parameters smaller than the threshold are masked to 0 upon convergence, and the optimizer is reset. For the KdV, KS, and nKdV equations, we set the threshold to 0.5, while for Burgers equation, we set it to $5 \times 10^{-3}$. All methods share the above experimental settings to ensure a fair comparison. We perform experiments on a single-core NVIDIA GeForce RTX 3090 GPU with available memory of $24{,}576$ MiB.

# F  ADDITIONAL EXPERIMENT

## F.1  ROBUSTNESS ANALYSIS AGAINST DATASET NOISE

Taking the Burgers equation as an example, we evaluate the robustness of DI-SINDy under increasing dataset noise levels. Note that, as mentioned in Appendix D, the main results of the Burgers equation in Section 5.2 are obtained under a dataset noise level of $\sigma = 10^{-3}$. Here, we further increase the dataset noise level to $\sigma = \{3 \times 10^{-3}, 5 \times 10^{-3}\}$. The corresponding success rates and RMSE of DI-SINDy and SINDy are shown in Table 4, while the long-term prediction errors are illustrated in Figure 4. It is noted that DI-SINDy ($\sigma = 10^{-3}$) exhibits a sufficiently small deviation from ground truth compared to other settings, such that their error curves almost overlap under the coordinate scale of Figure 4. As the dataset noise level increases, SINDy fails completely, whereas our DI-SINDy maintains a certain success rate and high accuracy, which demonstrates its stronger robustness against real-world disturbances.

Table 4: Success rates and RMSE of SINDy and DI-SINDy (Ours) for the Burgers equation with different dataset noise levels. All experimental results are averaged over 50 runs. RMSE is presented in the format of mean $\pm$ std.

| Dataset noise level | Method | Success rate ($\uparrow$) | RMSE (successful) ($\downarrow$) | RMSE (all) ($\downarrow$) |
|---|---|---|---|---|
| $\sigma = 10^{-3}$ | SINDy | $4\%$ | $(2.11 \pm 0.14) \times 10^{-2}$ | $(1.52 \pm 2.34) \times 10^{-1}$ |
| | DI-SINDy (Ours) | $\mathbf{98\%}$ | $\mathbf{(2.66 \pm 1.32) \times 10^{-4}}$ | $\mathbf{(4.02 \pm 9.62) \times 10^{-4}}$ |
| $\sigma = 3 \times 10^{-3}$ | SINDy | $0\%$ | N/A | $(3.13 \pm 2.23) \times 10^{-1}$ |
| | DI-SINDy (Ours) | $\mathbf{100\%}$ | $\mathbf{(2.14 \pm 0.50) \times 10^{-3}}$ | $\mathbf{(2.14 \pm 0.50) \times 10^{-3}}$ |
| $\sigma = 5 \times 10^{-3}$ | SINDy | $0\%$ | N/A | $(3.07 \pm 1.86) \times 10^{-1}$ |
| | DI-SINDy (Ours) | $\mathbf{28\%}$ | $\mathbf{(3.18 \pm 0.28) \times 10^{-3}}$ | $\mathbf{(5.98 \pm 1.75) \times 10^{-3}}$ |

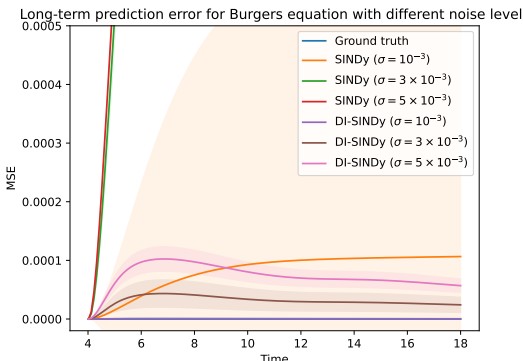

Figure 4: Long-term prediction errors of SINDy and DI-SINDy (Ours) for the Burgers equation with different dataset noise levels. The MSE at each time step is averaged over $4$ initial conditions and $50$ runs, with the shaded area representing the standard deviation.

### F.2 ROBUSTNESS ANALYSIS AGAINST INFINITESIMAL GENERATOR NOISE

Sometimes we cannot obtain a perfectly accurate symmetry group because our prior knowledge of the scientific problem is biased, or there are errors in the symmetry discovery method. Therefore, we further discuss the robustness of DI-SINDy with respect to noise in the infinitesimal generators. Taking the Burgers equation as an example again, we assume that the last infinitesimal generator is inaccurate: $\mathbf{v}_3 = (1 + \delta)t\frac{\partial}{\partial x} + \frac{\partial}{\partial u}$. Note that numerical perturbations on $\mathbf{v}_1 = (1+\delta)\frac{\partial}{\partial x}$ or $\mathbf{v}_2 = (1+\delta)\frac{\partial}{\partial t}$ do not affect the results, as they are equivalent as "bases" up to a constant factor. Then, similar to Equation (21), we obtain its prolongation:

$$\text{pr}^{(4)}\mathbf{v}_3 = (1+\delta)t\frac{\partial}{\partial x} + \frac{\partial}{\partial u} - (1+\delta)u_x\frac{\partial}{\partial u_t}. \tag{34}$$

The characteristic equation (23) becomes:

$$\mathrm{d}u = -\frac{\mathrm{d}u_t}{(1+\delta)u_x}. \tag{35}$$

Integrating, we obtain the biased differential invariants as $\{u_t + (1+\delta)uu_x, u_x, u_{xx}, u_{xxx}, u_{xxxx}\}$.

Table 5: Success rates and RMSE of different SINDy-based methods with varying infinitesimal generator noise levels for the Burgers equation. All experimental results are averaged over $50$ runs. RMSE is presented in the format of mean $\pm$ std.

| Method | Infinitesimal generator noise level | Success rate ($\uparrow$) | RMSE (successful) ($\downarrow$) | RMSE (all) ($\downarrow$) |
|---|---|---|---|---|
| SINDy | N/A | 4% | $(2.11 \pm 0.14) \times 10^{-2}$ | $(1.52 \pm 2.34) \times 10^{-1}$ |
| EquivSINDy-r ($\lambda = 10^{-3}$) | | 28% | $(8.89 \pm 2.52) \times 10^{-3}$ | $(1.30 \pm 3.66) \times 10^{-1}$ |
| EquivSINDy-r ($\lambda = 10^{-2}$) | $\delta = 10^{-1}$ | 58% | $(4.54 \pm 0.66) \times 10^{-2}$ | $(1.37 \pm 3.77) \times 10^{-1}$ |
| EquivSINDy-r ($\lambda = 10^{-1}$) | | 82% | $(6.75 \pm 0.07) \times 10^{-2}$ | $(1.61 \pm 3.79) \times 10^{-1}$ |
| DI-SINDy (Ours) | | **100**% | $(\mathbf{2.16 \pm 1.16}) \times \mathbf{10^{-4}}$ | $(\mathbf{2.16 \pm 1.16}) \times \mathbf{10^{-4}}$ |
| EquivSINDy-r ($\lambda = 10^{-3}$) | | 36% | $(3.08 \pm 1.22) \times 10^{-2}$ | $(1.94 \pm 4.04) \times 10^{-1}$ |
| EquivSINDy-r ($\lambda = 10^{-2}$) | $\delta = 3 \times 10^{-1}$ | 40% | $(1.51 \pm 0.11) \times 10^{-1}$ | $(1.75 \pm 1.18) \times 10^{-1}$ |
| EquivSINDy-r ($\lambda = 10^{-1}$) | | 64% | $(2.05 \pm 0.02) \times 10^{-1}$ | $(2.82 \pm 3.45) \times 10^{-1}$ |
| DI-SINDy (Ours) | | **90**% | $(\mathbf{1.08 \pm 0.24}) \times \mathbf{10^{-3}}$ | $(\mathbf{1.09 \pm 0.24}) \times \mathbf{10^{-3}}$ |
| EquivSINDy-r ($\lambda = 10^{-3}$) | | 22% | $(5.35 \pm 1.33) \times 10^{-2}$ | $(2.19 \pm 4.04) \times 10^{-1}$ |
| EquivSINDy-r ($\lambda = 10^{-2}$) | $\delta = 5 \times 10^{-1}$ | 22% | $(2.54 \pm 0.24) \times 10^{-1}$ | $(2.87 \pm 3.15) \times 10^{-1}$ |
| EquivSINDy-r ($\lambda = 10^{-1}$) | | 54% | $(3.42 \pm 0.04) \times 10^{-1}$ | $(3.92 \pm 2.99) \times 10^{-1}$ |
| DI-SINDy (Ours) | | **70**% | $(\mathbf{1.94 \pm 0.35}) \times \mathbf{10^{-3}}$ | $(\mathbf{1.99 \pm 0.33}) \times \mathbf{10^{-3}}$ |

In practice, we set different infinitesimal generator noise levels $\delta = \{10^{-1}, 3 \times 10^{-1}, 5 \times 10^{-1}\}$ for EquivSINDy-r and DI-SINDy, respectively, for evaluation. The corresponding success rates and RMSE are presented in Table 5. We note that sometimes EquivSINDy-r with a smaller regularization weight $\lambda$ exhibits an increase rather than a decrease in success rate as $\delta$ increases. This is

because it inherently struggles to filter out key terms from the large search space, while the enhanced correlation coefficients of the infinitesimal generators highlight the correct answer. Overall, in most cases, the accuracy of symmetry-informed equation discovery methods decreases as the perturbation magnitude increases, but our DI-SINDy consistently performs better.

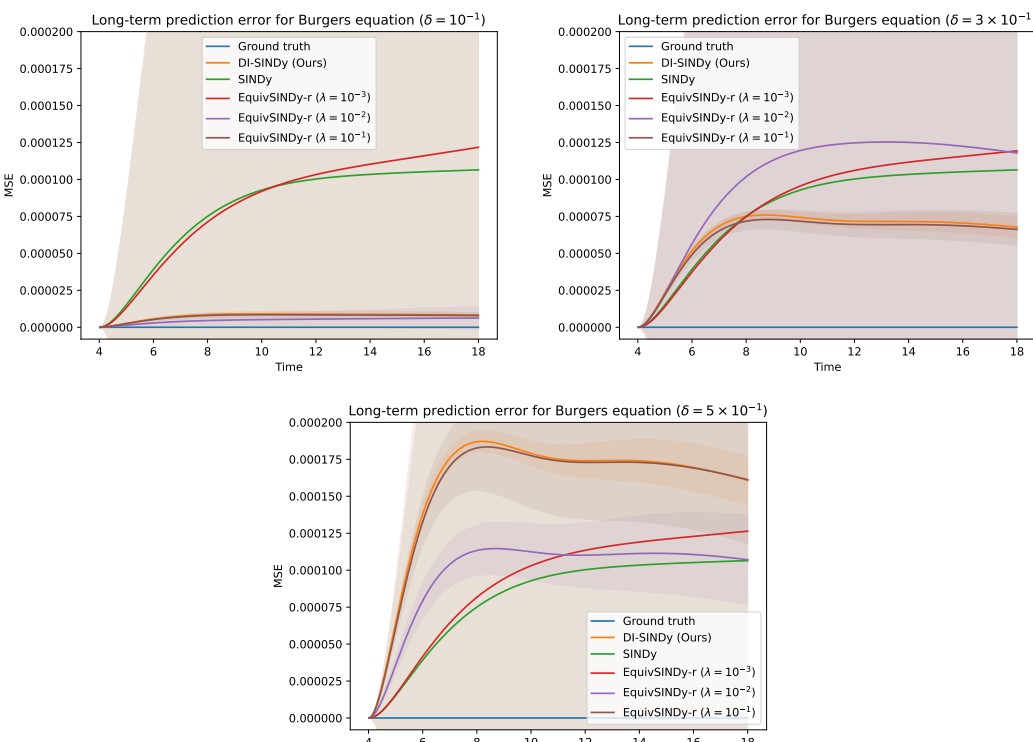

Figure 5: Long-term prediction errors of different SINDy-based methods with varying infinitesimal generator noise levels for the Burgers equation. The MSE at each time step is averaged over 4 initial conditions and 50 runs, with the shaded area representing the standard deviation.

Figure 5 illustrates the variation of long-term prediction errors with respect to $\delta$ for different SINDy-based methods. We observe that the results of DI-SINDy and EquivSINDy-r with larger $\lambda$ are more affected by errors in the infinitesimal generator—this is theoretically reasonable, as they adhere more strictly to symmetry. Even so, compared to SINDy, which does not incorporate symmetry information, our DI-SINDy achieves more stable long-term prediction results as long as $\delta$ does not exceed $3 \times 10^{-1}$. This demonstrates that DI-SINDy has a high tolerance for perturbations in the infinitesimal generator, indicating its strong potential for integration with symmetry discovery methods.

### F.3 HIGH-DIMENSIONAL CASE: GOVERNING EQUATION DISCOVERY FOR REACTION-DIFFUSION SYSTEM

Furthermore, we attempt to solve the reaction-diffusion system (Champion et al., 2019) using our method, for which the explicit equation is:

$$\begin{cases} u_t = (1 - (u^2 + v^2))u + \beta(u^2 + v^2)v + d_1(u_{xx} + u_{yy}), \\ v_t = -\beta(u^2 + v^2)u + (1 - (u^2 + v^2))v + d_2(v_{xx} + v_{yy}), \end{cases} \tag{36}$$

with $d_1 = 0.1$, $d_2 = 0.1$, and $\beta = 1$. This is a high-dimensional case where both the independent variables $(t, x, y) \in X$ and the dependent variables $(u, v) \in U$ are vectors. The initial terms for its second-order prolongation are $\{t, x, y, u, v, u_x, u_y, v_x, v_y, u_{xx}, u_{xy}, u_{yy}, v_{xx}, v_{xy}, v_{yy}, u_t, v_t\}$. This constitutes a very large search space, and according to the form of Equation (36), we need to extend it to at least third-order terms to include the correct solution, which would undoubtedly overwhelm

methods such as SINDy and EquivSINDy-r (as mentioned in the original paper (Yang et al., 2024b), EquivSINDy-r only attempts to operate in the latent space). We will next demonstrate how to address this challenge based on differential invariants.

According to (Hu et al., 2025a), we know that this system possesses spatiotemporal translation and $SO(2)$ symmetry, with the corresponding infinitesimal generators being:

$$\mathbf{v}_1 = -v\frac{\partial}{\partial u} + u\frac{\partial}{\partial v}, \quad \mathbf{v}_2 = -y\frac{\partial}{\partial x} + x\frac{\partial}{\partial y}, \quad \mathbf{v}_3 = \frac{\partial}{\partial t}, \quad \mathbf{v}_4 = \frac{\partial}{\partial x}, \quad \mathbf{v}_5 = \frac{\partial}{\partial y}, \quad (37)$$

where $\{\mathbf{v}_3, \mathbf{v}_4, \mathbf{v}_5\}$ can narrow the search space to $\{u, v, u_x, u_y, v_x, v_y, u_{xx}, u_{xy}, u_{yy}, v_{xx}, v_{xy}, v_{yy}, u_t, v_t\}$. Then we calculate the second-order prolongation of $\{\mathbf{v}_1, \mathbf{v}_2\}$:

$$\begin{cases} \mathrm{pr}^{(2)}\mathbf{v}_1 = & (-v\frac{\partial}{\partial u} + u\frac{\partial}{\partial v}) + (-v_t\frac{\partial}{\partial u_t} + u_t\frac{\partial}{\partial v_t}) + (-v_x\frac{\partial}{\partial u_x} + u_x\frac{\partial}{\partial v_x}) + (-v_y\frac{\partial}{\partial u_y} + u_y\frac{\partial}{\partial v_y}) \\ & + (-v_{xx}\frac{\partial}{\partial u_{xx}} + u_{xx}\frac{\partial}{\partial v_{xx}}) + (-v_{xy}\frac{\partial}{\partial u_{xy}} + u_{xy}\frac{\partial}{\partial v_{xy}}) + (-v_{yy}\frac{\partial}{\partial u_{yy}} + u_{yy}\frac{\partial}{\partial v_{yy}}), \\ \mathrm{pr}^{(2)}\mathbf{v}_2 = & (-y\frac{\partial}{\partial x} + x\frac{\partial}{\partial y}) + (-u_y\frac{\partial}{\partial u_x} + u_x\frac{\partial}{\partial u_y}) + (-v_y\frac{\partial}{\partial v_x} + v_x\frac{\partial}{\partial v_y}) \\ & + (-2u_{xy}\frac{\partial}{\partial u_{xx}} + 2u_{xy}\frac{\partial}{\partial u_{yy}}) + (-2v_{xy}\frac{\partial}{\partial v_{xx}} + 2v_{xy}\frac{\partial}{\partial v_{yy}}) \\ & + (u_{xx} - u_{yy})\frac{\partial}{\partial u_{xy}} + (v_{xx} - v_{yy})\frac{\partial}{\partial v_{xy}}. \end{cases} \quad (38)$$

Construct the characteristic equation based on $\mathrm{pr}^{(2)}\mathbf{v}_1$:

$$\frac{\mathrm{d}v}{u} = -\frac{\mathrm{d}u}{v} = \frac{\mathrm{d}v_x}{u_x} = -\frac{\mathrm{d}u_x}{v_x} = \frac{\mathrm{d}v_y}{u_y} = -\frac{\mathrm{d}u_y}{v_y} = \frac{\mathrm{d}v_t}{u_t} = -\frac{\mathrm{d}u_t}{v_t}$$

$$= \frac{\mathrm{d}v_{xx}}{u_{xx}} = -\frac{\mathrm{d}u_{xx}}{v_{xx}} = \frac{\mathrm{d}v_{xy}}{u_{xy}} = -\frac{\mathrm{d}u_{xy}}{v_{xy}} = \frac{\mathrm{d}v_{yy}}{u_{yy}} = -\frac{\mathrm{d}u_{yy}}{v_{yy}}. \quad (39)$$

The search space is then updated to $\{u^2 + v^2, vu_x - uv_x, uu_x + vv_x, vu_y - uv_y, uu_y + vv_y, vu_{xx} - uv_{xx}, uu_{xx} + vv_{xx}, vu_{xy} - uv_{xy}, uu_{xy} + vv_{xy}, vu_{yy} - uv_{yy}, uu_{yy} + vv_{yy}, vu_t - uv_t, uu_t + vv_t\}$. These terms transform under $\mathrm{pr}^{(2)}\mathbf{v}_2$ as:

$$\mathrm{pr}^{(2)}\mathbf{v}_2 \begin{bmatrix} u^2 + v^2 \\ vu_x - uv_x \\ uu_x + vv_x \\ vu_y - uv_y \\ uu_y + vv_y \\ vu_{xx} - uv_{xx} \\ uu_{xx} + vv_{xx} \\ vu_{xy} - uv_{xy} \\ uu_{xy} + vv_{xy} \\ vu_{yy} - uv_{yy} \\ uu_{yy} + vv_{yy} \\ vu_t - uv_t \\ uu_t + vv_t \end{bmatrix} = \begin{bmatrix} 0 \\ -vu_y + uv_y \\ -uu_y - vv_y \\ vu_x - uv_x \\ uu_x + vv_x \\ -2vu_{xy} + 2uv_{xy} \\ -2uu_{xy} - 2vv_{xy} \\ v(u_{xx} - u_{yy}) - u(v_{xx} - v_{yy}) \\ u(u_{xx} - u_{yy}) + v(v_{xx} - v_{yy}) \\ 2vu_{xy} - 2uv_{xy} \\ 2uu_{xy} + 2vv_{xy} \\ 0 \\ 0 \end{bmatrix}. \quad (40)$$

To ensure invariance under $\mathrm{pr}^{(2)}\mathbf{v}_2$, we obtain the final differential invariants as $\{u^2 + v^2, (vu_{xx} - uv_{xx}) + (vu_{yy} - uv_{yy}), (uu_{xx} + vv_{xx}) + (uu_{yy} + vv_{yy}), vu_t - uv_t, uu_t + vv_t\}$.

After all this, we found that the set of differential invariants is extremely concise compared to the original search space. Based on this, DI-SINDy can construct the equation skeleton:

$$\begin{cases} vu_t - uv_t = W_1\Theta_1(u^2 + v^2, vu_{xx} - uv_{xx} + vu_{yy} - uv_{yy}, uu_{xx} + vv_{xx} + uu_{yy} + vv_{yy}), \\ uu_t + vv_t = W_2\Theta_2(u^2 + v^2, vu_{xx} - uv_{xx} + vu_{yy} - uv_{yy}, uu_{xx} + vv_{xx} + uu_{yy} + vv_{yy}). \end{cases} \quad (41)$$

In practice, we specify that $\Theta_1$ and $\Theta_2$ include terms up to second order, so the number of their terms is 9. On the other hand, if a function library is constructed based on the original search space while including terms up to third order, a rough estimate suggests it would contain hundreds of terms! Therefore, our DI-SINDy is theoretically more effective in handling such high-dimensional scenarios.

We conduct experiments using the dataset constructed in Hu et al. (2025a). The sparse regression threshold is set to $5 \times 10^{-2}$, while other settings remain consistent with the main experiments.

Since SINDy and EquivSINDy-r break down as mentioned earlier, we only analyze the results of DI-SINDy individually. We sample 10 discovered equations in Table 6. By observing their forms, we can roughly summarize the results as:

$$\begin{cases} vu_t - uv_t = 0.1(vu_{xx} - uv_{xx} + vu_{yy} - uv_{yy}) + (u^2 + v^2)^2, \\ uu_t + vv_t = (u^2 + v^2) + 0.1(uu_{xx} + vv_{xx} + uu_{yy} + vv_{yy}) - (u^2 + v^2)^2. \end{cases} \tag{42}$$

Equivalently:

$$\begin{cases} u_t = (1 - (u^2 + v^2))u + (u^2 + v^2)v + 0.1(u_{xx} + u_{yy}), \\ v_t = -(u^2 + v^2)u + (1 - (u^2 + v^2))v + 0.1(v_{xx} + v_{yy}). \end{cases} \tag{43}$$

Qualitatively, our DI-SINDy has discovered the correct form of the equation! We further present the quantitative results of 50 runs in Table 7. As shown in Table 6, the failure cases are all due to minor coefficient differences in $(u^2 + v^2)$ caused by $(u^2 + v^2)^2$, but the main forms remain correct. Overall, our DI-SINDy demonstrates significant potential for discovering complex PDEs in high-dimensional dynamic systems.

Table 6: Samples of equation discovery results by DI-SINDy for the reaction-diffusion system.

| # | Discovered equation |
|---|---|
| 1 | $vu_t - uv_t = -0.0547(u^2 + v^2) + 0.1047(vu_{xx} - uv_{xx} + vu_{yy} - uv_{yy}) + 1.0136(u^2 + v^2)^2$ 
 $uu_t + vv_t = +1.0176(u^2 + v^2) + 0.1065(uu_{xx} + vv_{xx} + uu_{yy} + vv_{yy}) - 1.0036(u^2 + v^2)^2$ |
| 2 | $vu_t - uv_t = +0.1059(vu_{xx} - uv_{xx} + vu_{yy} - uv_{yy}) + 1.0088(u^2 + v^2)^2$ 
 $uu_t + vv_t = +1.0157(u^2 + v^2) + 0.1059(uu_{xx} + vv_{xx} + uu_{yy} + vv_{yy}) - 1.0033(u^2 + v^2)^2$ |
| 3 | $vu_t - uv_t = -0.0602(u^2 + v^2) + 0.1044(vu_{xx} - uv_{xx} + vu_{yy} - uv_{yy}) + 1.0148(u^2 + v^2)^2$ 
 $uu_t + vv_t = +1.0190(u^2 + v^2) + 0.1065(uu_{xx} + vv_{xx} + uu_{yy} + vv_{yy}) - 1.0039(u^2 + v^2)^2$ |
| 4 | $vu_t - uv_t = +0.1062(vu_{xx} - uv_{xx} + vu_{yy} - uv_{yy}) + 1.0089(u^2 + v^2)^2$ 
 $uu_t + vv_t = +1.0158(u^2 + v^2) + 0.1058(uu_{xx} + vv_{xx} + uu_{yy} + vv_{yy}) - 1.0033(u^2 + v^2)^2$ |
| 5 | $vu_t - uv_t = -0.0526(u^2 + v^2) + 0.1039(vu_{xx} - uv_{xx} + vu_{yy} - uv_{yy}) + 1.0130(u^2 + v^2)^2$ 
 $uu_t + vv_t = +1.0167(u^2 + v^2) + 0.1062(uu_{xx} + vv_{xx} + uu_{yy} + vv_{yy}) - 1.0034(u^2 + v^2)^2$ |
| 6 | $vu_t - uv_t = -0.0607(u^2 + v^2) + 0.1048(vu_{xx} - uv_{xx} + vu_{yy} - uv_{yy}) + 1.0148(u^2 + v^2)^2$ 
 $uu_t + vv_t = +1.0191(u^2 + v^2) + 0.1066(uu_{xx} + vv_{xx} + uu_{yy} + vv_{yy}) - 1.0039(u^2 + v^2)^2$ |
| 7 | $vu_t - uv_t = -0.0551(u^2 + v^2) + 0.1037(vu_{xx} - uv_{xx} + vu_{yy} - uv_{yy}) + 1.0136(u^2 + v^2)^2$ 
 $uu_t + vv_t = +1.0175(u^2 + v^2) + 0.1064(uu_{xx} + vv_{xx} + uu_{yy} + vv_{yy}) - 1.0036(u^2 + v^2)^2$ |
| 8 | $vu_t - uv_t = +0.1052(vu_{xx} - uv_{xx} + vu_{yy} - uv_{yy}) + 1.0079(u^2 + v^2)^2$ 
 $uu_t + vv_t = +1.0133(u^2 + v^2) + 0.1052(uu_{xx} + vv_{xx} + uu_{yy} + vv_{yy}) - 1.0029(u^2 + v^2)^2$ |
| 9 | $vu_t - uv_t = -0.0558(u^2 + v^2) + 0.1046(vu_{xx} - uv_{xx} + vu_{yy} - uv_{yy}) + 1.0138(u^2 + v^2)^2$ 
 $uu_t + vv_t = +1.0179(u^2 + v^2) + 0.1065(uu_{xx} + vv_{xx} + uu_{yy} + vv_{yy}) - 1.0036(u^2 + v^2)^2$ |
| 10 | $vu_t - uv_t = +0.1058(vu_{xx} - uv_{xx} + vu_{yy} - uv_{yy}) + 1.0075(u^2 + v^2)^2$ 
 $uu_t + vv_t = +1.0127(u^2 + v^2) + 0.1051(uu_{xx} + vv_{xx} + uu_{yy} + vv_{yy}) - 1.0028(u^2 + v^2)^2$ |

Table 7: Success rate and RMSE of DI-SINDy for the reaction-diffusion system. Experimental results are averaged over 50 runs. RMSE is presented in the format of mean $\pm$ std.

| Success rate ($\uparrow$) | RMSE (successful) ($\downarrow$) | RMSE (all) ($\downarrow$) |
|---|---|---|
| 48% | $(8.35 \pm 0.59) \times 10^{-3}$ | $(9.53 \pm 1.26) \times 10^{-3}$ |

## F.4 DIFFERENTIAL INVARIANTS GUIDE TRANSFORMER-BASED GOVERNING EQUATION DISCOVERY

Differential invariants can be plug-and-play with existing equation discovery methods beyond SINDy. Now, we further take E2E (Kamienny et al., 2022)—a transformer-based symbolic regression method—as an example to demonstrate the strong versatility of our pipeline. Similar to DI-SINDy, we incorporate differential invariants as relevant terms into the pre-trained E2E model, referring to it as E2E based on Differential Invariants (DI-E2E). Both E2E and DI-E2E infer an explicit expression $F$ from a set of input data points. Note that $F$ is represented in the form of an expression tree without a fixed skeleton, so the relevant metrics of the SINDy-based methods, such as success rate and RMSE of the coefficient matrix, are not applicable. We adopt the $R^2$**-score** and **accuracy to tolerance** $\tau$ from the original paper (Kamienny et al., 2022) to evaluate E2E and DI-E2E, which are defined as follows:

$$R^2 = 1 - \frac{\sum_{i=1}^{N_{\text{test}}} \|y_i - F(x_i)\|^2}{\sum_{i=1}^{N_{\text{test}}} \|y_i - \bar{y}\|^2}, \quad \text{Acc}_\tau = \frac{1}{N_{\text{test}}} \sum_{i=1}^{N_{\text{test}}} \mathbb{1}\left(\frac{\|F(x_i) - y_i\|}{\|y_i\|} \le \tau\right), \tag{44}$$

where $\mathbb{1}$ is the indicator function, and $\mathcal{D}_{\text{test}} = \{(x_i, y_i)\}_{i=1}^{N_{\text{test}}}$ is the test dataset.

Table 8: $R^2$-score, accuracy to tolerance $\tau$, and inference time of E2E and DI-E2E for the KdV, KS, Burgers, nKdV, and reaction-diffusion (RD) equations. All experimental results are averaged over 50 runs and presented in the format of mean $\pm$ std.

| Name | Method | $R^2$ ($\uparrow$) | $\text{Acc}_{0.1}$($\uparrow$) | $\text{Acc}_{0.01}$($\uparrow$) | $\text{Acc}_{0.001}$($\uparrow$) | Inference time (s) ($\downarrow$) |
|---|---|---|---|---|---|---|
| KdV | E2E | $(5.07 \pm 3.12) \times 10^{-1}$ | $(7.01 \pm 6.43) \times 10^{-2}$ | $(6.57 \pm 5.40) \times 10^{-3}$ | $(6.35 \pm 5.37) \times 10^{-4}$ | $120 \pm 4$ |
| | DI-E2E (Ours) | $(\mathbf{5.51 \pm 3.19}) \times 10^{-1}$ | $(\mathbf{1.04 \pm 0.81}) \times 10^{-1}$ | $(\mathbf{1.13 \pm 1.38}) \times 10^{-2}$ | $(\mathbf{1.08 \pm 1.17}) \times 10^{-3}$ | $\mathbf{115 \pm 5}$ |
| KS | E2E | $(7.37 \pm 12.85) \times 10^{-2}$ | $(2.15 \pm 1.79) \times 10^{-2}$ | $(2.19 \pm 1.86) \times 10^{-3}$ | $(2.29 \pm 2.87) \times 10^{-4}$ | $131 \pm 6$ |
| | DI-E2E (Ours) | $(\mathbf{4.40 \pm 2.38}) \times 10^{-1}$ | $(\mathbf{8.80 \pm 2.90}) \times 10^{-2}$ | $(\mathbf{9.02 \pm 3.66}) \times 10^{-3}$ | $(\mathbf{8.47 \pm 4.02}) \times 10^{-4}$ | $\mathbf{95.7 \pm 1.1}$ |
| Burgers | E2E | $(9.94 \pm 0.10) \times 10^{-1}$ | $(7.95 \pm 1.48) \times 10^{-1}$ | $(1.25 \pm 0.76) \times 10^{-1}$ | $(1.26 \pm 1.06) \times 10^{-2}$ | $109 \pm 3$ |
| | DI-E2E (Ours) | $(\mathbf{9.97 \pm 0.03}) \times 10^{-1}$ | $(\mathbf{8.65 \pm 0.68}) \times 10^{-1}$ | $(\mathbf{2.92 \pm 1.62}) \times 10^{-1}$ | $(\mathbf{3.25 \pm 4.47}) \times 10^{-2}$ | $\mathbf{102 \pm 1}$ |
| nKdV | E2E | $(2.09 \pm 2.52) \times 10^{-1}$ | $(3.47 \pm 2.34) \times 10^{-2}$ | $(3.50 \pm 2.54) \times 10^{-3}$ | $(3.17 \pm 3.13) \times 10^{-4}$ | $121 \pm 3$ |
| | DI-E2E (Ours) | $(\mathbf{2.77 \pm 3.17}) \times 10^{-1}$ | $(\mathbf{7.72 \pm 7.08}) \times 10^{-2}$ | $(\mathbf{7.03 \pm 5.56}) \times 10^{-3}$ | $(\mathbf{7.57 \pm 6.67}) \times 10^{-4}$ | $\mathbf{103 \pm 2}$ |
| RD | E2E | NaN | NaN | NaN | NaN | $35.0 \pm 0.9$ |
| | DI-E2E (Ours) | $(\mathbf{9.99 \pm 0.00}) \times 10^{-1}$ | $(\mathbf{9.62 \pm 0.13}) \times 10^{-1}$ | $(\mathbf{7.04 \pm 2.51}) \times 10^{-1}$ | $(\mathbf{1.94 \pm 3.71}) \times 10^{-2}$ | $\mathbf{20.2 \pm 1.0}$ |

In Table 8, we present the quantitative evaluation results of E2E and DI-E2E for the KdV, KS, Burgers, nKdV, and reaction-diffusion equations, along with the corresponding inference times. The settings for all datasets remain consistent with those in Appendix D. Specifically, for the reaction-diffusion system, due to its vast search space, we sample only $8,192$ data points to ensure efficient inference. Although E2E generates more flexible expressions compared to SINDy, its overall performance is unstable due to noise in the dataset and errors introduced by central differencing. However, our DI-E2E demonstrates significant improvements across all metrics while requiring less computational time. Notably, E2E completely fails for the high-dimensional and complex reaction-diffusion system, whereas our DI-E2E achieves an $R^2$-score close to 1, which is nearly perfect.

## G   LLM Usage

Large language models are only used for writing polishing, including word spelling checks, grammar error checks, translation, and so on.

