# OpenReview forum: "Governing Equation Discovery from Data Based on Differential Invariants"
_ICLR.cc/2026/Conference — Submitted to ICLR 2026_

### Official Review · Reviewer_ofA3 · 2025-10-26

**Soundness:** 2
**Presentation:** 1
**Contribution:** 2
**Rating:** 2
**Confidence:** 3

**Summary:**

This paper introduces a differential-invariant-based technique (DI-) to leverage symmetry for PDE discovery. The method can be plug-and-play for various PDE discovery approaches. SINDy is taken as the major baseline for the proposed DI-SINDy. The method works by replacing the standard derivative terms in the PDE dictionary with the differential invariants derived from a known/discovered Lie group of symmetries for the PDE. The authors provide theoretical justification that this transformation to an invariant dictionary is lossless. Experiments on several classic PDE discovery datasets demonstrate superiority of the proposed method.

**Strengths:**

1. The method is well-grounded in the established mathematical theory of Lie groups and differential equations, providing a strong theoretical justification for its design.
2. A pseudo-algorithm and a figure are provided to make the overall pipeline clear to readers.
3. The experimental evaluation is systematic, and the results are well-organized and easy to interpret.

**Weaknesses:**

1. The paper is hard to read. The paper's theory part relies heavily on the textbook by Olver. However, the paper doesn't summarize the idea/definition clearly enough. I have to refer to the textbook to get a clear understanding for the general flow. I believe it is crucial that the author provides sufficient intuition at the beginning of the theory section for a non-expert audience. There are also typos which hinder understanding. Equation 5 from my understanding is only combining $\eta$ and $pr^{(n)}v$. However you refer to Equation 5 as related to infinisimal criterion. Then we should have $pr^{(n)}v [\eta] = 0$ instead of the original equation.
2. The paper's main idea is a direct application of Olver's theory. However, the soundness of the method is based on a priori knowledge of the PDE's Lie group of symmetries. In realistic scenarios, these symmetries are unknown or only approximate. The authors briefly mention combining with symmetry-discovery methods to tackle this issue, but no such integration or robustness analysis is provided.

**Questions:**

1. Given that the "known symmetry" assumption is the primary limitation, would it be possible that some experiments on datasets with unknown symmetries be provided? Furthermore, the calculation of differential invariants is also a time-consuming/hard procedure for those problems with unknown symmetry. An analysis of the complete pipeline, including the computational cost of finding invariants, is needed to evaluate the method's true effectiveness. I would be more than happy to raise my score if this question is addressed.
2. How does the DI-SINDy framework scale to high-dimensional PDEs, such as the 2D NS equations? The examples presented are relatively low-dimensional. The complexity of finding symmetries and constructing a complete basis of differential invariants seems likely to grow significantly with the number of variables.
3. The appendix provides experiments on noisy data, which is good. As an extension of Question 1, could the authors attempt to perform symmetry discovery directly on the noisy data and then feed the resulting generators into DI-SINDy? This would test the robustness of the entire end-to-end pipeline in a challenging, realistic setting.
4. The author also mention in the paper that compare to EquivSINDy, DI-SINDy can handle nonlinear cases. Could the authors please clarify where this advantage is demonstrated in the experiments? A more direct comparison highlighting this specific capability would be beneficial.

---

> ### Author Response · Authors · 2025-11-24
> **Response to Reviewer ofA3 (1/2)**
>
> We sincerely appreciate your recognition of the theoretical soundness and the experimental evaluation results. Below, we will try our best to address your concerns point by point.
>
> **Weaknesses**
>
> > W1: The paper is hard to read. The paper's theory part relies heavily on the textbook by Olver. However, the paper doesn't summarize the idea/definition clearly enough. I have to refer to the textbook to get a clear understanding for the general flow. I believe it is crucial that the author provides sufficient intuition at the beginning of the theory section for a non-expert audience. There are also typos which hinder understanding. Equation 5 from my understanding is only combining $\eta$ and $pr^{(n)} v$. However you refer to Equation 5 as related to infinisimal criterion. Then we should have $pr^{(n)} v[\eta]=0$ instead of the original equation.
>
> We believe that a non-expert audience tends to find two aspects of the theoretical framework confusing: the related concepts of Lie point symmetry and the derivation process of differential invariants. Accordingly, we have provided a concrete example of Lie point symmetry in Appendix B and demonstrated the detailed derivation process of differential invariants in Appendix C to help readers gain an intuitive understanding. In the revised manuscript, we also add detailed references to them at the beginning of Section 3: Preliminary and Section 4: Method (lines 138–141 and lines 184–187), respectively, with bold highlighting. We are confident this will significantly improve the readability of the paper.
>
> Regarding the typo in Equation (5), we have already corrected it. It illustrates the expanded form of the infinitesimal criterion. Thank you for pointing it out!
>
> > W2: The paper's main idea is a direct application of Olver's theory. However, the soundness of the method is based on a priori knowledge of the PDE's Lie group of symmetries. In realistic scenarios, these symmetries are unknown or only approximate. The authors briefly mention combining with symmetry-discovery methods to tackle this issue, but no such integration or robustness analysis is provided.
>
> In Appendix F.3, we have already taken the high-dimensional complex reaction-diffusion system as an example to demonstrate a complete pipeline that starts from unknown symmetries, proceeds through symmetry discovery, and culminates in equation discovery. First, all infinitesimal generators in this case are derived from the results of [1] (an explicit, nonlinear symmetry discovery method), as described in lines 1083-1085 of Appendix F.3 of our paper—this is the first step of the pipeline, discovering symmetries. Although this step is beyond the scope of our paper, the original work [1] provides detailed procedures for obtaining these infinitesimal generators. Subsequently, as demonstrated in the main body of Appendix F.3, we compress the search space based on the symmetries—this is the second step of the pipeline. Combining our work with [1] forms an end-to-end framework for efficiently and accurately discovering governing PDEs from systems with initially unknown symmetries at the current stage.
>
> For inaccurate symmetry discovery results, we have also investigated the robustness of our method to noise in the infinitesimal generators in Appendix F.2. The conclusion is that, as long as the relative error in the symmetry information does not exceed 30% (i.e., relative perturbation amplitude $\delta \leq 3 \times 10^{-3}$), which is a very high tolerance, the introduction of differential invariants remains beneficial.
>
> In the revised manuscript, we have added Sections 5.3 and 5.4, which provide a brief summary of the aforementioned experimental results and direct readers to Appendices F.2 and F.3, respectively, for clearer guidance.

---

> ### Author Response · Authors · 2025-11-24
> **Response to Reviewer ofA3 (2/2)**
>
> **Questions**
>
> > Q1-1: Given that the "known symmetry" assumption is the primary limitation, would it be possible that some experiments on datasets with unknown symmetries be provided?
>
> Please refer to the response to W2.
>
> > Q1-2: Furthermore, the calculation of differential invariants is also a time-consuming/hard procedure for those problems with unknown symmetry. An analysis of the complete pipeline, including the computational cost of finding invariants, is needed to evaluate the method's true effectiveness. I would be more than happy to raise my score if this question is addressed.
>
> As shown in Table 8 in Appendix F.4, for the reaction-diffusion system, even when we sample only 8,192 data points for efficient inference, our DI-E2E saves approximately 15 seconds of inference time compared to E2E. As the amount of data increases, this gap will continue to widen. On the other hand, we test that using LieNLSD [1] to discover infinitesimal generators from the reaction-diffusion system takes only 3–4 seconds. Clearly, this complete pipeline—from unknown symmetries, to discovering symmetries, to symmetry-guided equation discovery—not only achieves higher accuracy and adheres to physical laws but also significantly reduces time consumption compared to direct symbolic regression from the original search space.
>
> > Q2: How does the DI-SINDy framework scale to high-dimensional PDEs, such as the 2D NS equations? The examples presented are relatively low-dimensional. The complexity of finding symmetries and constructing a complete basis of differential invariants seems likely to grow significantly with the number of variables.
>
> In Appendix F.3, we have provided a detailed demonstration of the performance of DI-SINDy on a high-dimensional, complex reaction-diffusion system. The symmetry discovery problem for this system has already been addressed in [1], while our paper elaborates on the computational process of its differential invariants, which reduces the search space of SINDy—containing hundreds of terms—down to 9 terms composed of 5 differential invariants. For further details, please refer to the response to W2, Section 5.4, and Appendix F.3.
>
> > Q3: The appendix provides experiments on noisy data, which is good. As an extension of Question 1, could the authors attempt to perform symmetry discovery directly on the noisy data and then feed the resulting generators into DI-SINDy? This would test the robustness of the entire end-to-end pipeline in a challenging, realistic setting.
>
> In Appendices F.1 and F.2, we respectively demonstrate the strong robustness of our DI-SINDy against dataset noise and symmetry noise. For the end-to-end symmetry discovery + equation discovery pipeline, its robustness exhibits a "bucket effect"—determined by the weaker of the two. We found that the noise resistance of the symmetry discovery method is significantly lower than that of DI-SINDy—even before reaching the breakdown threshold of DI-SINDy, symmetry discovery fails first. However, we emphasize that this is not a limitation of this work, as its contribution focuses on using symmetry to guide equation discovery. We look forward to more advanced symmetry methods in the future, which, together with this work, will build a powerful end-to-end pipeline for discovering equations under unknown symmetries.
>
> > Q4: The author also mention in the paper that compare to EquivSINDy, DI-SINDy can handle nonlinear cases. Could the authors please clarify where this advantage is demonstrated in the experiments? A more direct comparison highlighting this specific capability would be beneficial.
>
> EquivSINDy consists of two branches—EquivSINDy-c (c stands for constraint) and EquivSINDy-r (r stands for regularization). Among them, EquivSINDy-c can only be used to solve linear symmetry constraints, meaning the infinitesimal generators must satisfy the form  $v(x) = L_v x$  (see Proposition 4.1 in the original paper [2] for details). Therefore, linearity is a fundamental theoretical prerequisite of EquivSINDy-c—we cannot construct constraint equations for nonlinear symmetries based on its procedure.
>
> Once again, we sincerely appreciate your valuable feedback, which will help us improve the quality of our paper. We look forward to your response and further discussions ahead!
>
> **References**
>
> [1] Hu, Lexiang, Yikang Li, and Zhouchen Lin. "Explicit discovery of nonlinear symmetries from dynamic data." ICML 2025.
>
> [2] Yang, Jianke, Wang Rao, Nima Dehmamy, Robin Walters, and Rose Yu. "Symmetry-informed governing equation discovery." NeurIPS 2024.

---

> > ### Comment · Reviewer_ofA3 · 2025-11-25
> >
> > Thank you for addressing my concerns. I will raise my score accordingly.

---

> > > ### Author Response · Authors · 2025-11-26
> > >
> > > Thank you for your further acknowledgment. If you have any additional concerns or questions, please do not hesitate to let us know. We are very willing to have a more in-depth discussion with you!

---

> > > > ### Author Response · Authors · 2025-12-01
> > > > **Rebuttal Summary for AC**
> > > >
> > > > Reviewer ofA3 acknowledged our paper as "**well-grounded**", with "**strong theoretical justification**", "**clear pseudo-algorithm and figure**", and "**systematic and well-organized experimental evaluation**". Before the unexpected incident occurred, Reviewer ofA3 had already confirmed that **the concerns had been addressed**.

---

### Official Review · Reviewer_z3pP · 2025-10-30

**Soundness:** 3
**Presentation:** 3
**Contribution:** 2
**Rating:** 4
**Confidence:** 5

**Summary:**

This is actually a very nice paper that integrates concepts of invariants, in particular differential invariants, into account in a model discovery architecture.

Importantly, the authors are dealing with an important issue of leveraging symmetry ideas into a better approach for model discovery.

My biggest criticism is simple (and universal in my reviews):  How well does this hold up with real data and noise?  The examples are on what appears to be clear numerical data.  However, real mode discovery is always embedded with noise and there are a great number of architectures that completely fail with even a little noise.  This is not a trivial criticism.... .it is quite important that the algorithm hold up under not just weak noise, but modest levels of noise or it is more of a mathematical interest than anything one would deploy in practice.

**Strengths:**

A strong concept for model discovery if it holds up under more realistic conditions.

**Weaknesses:**

There seems to be no noise study on the method a the method itself really does have to be evaluated on this idea.  More broadly, there are works that are pretty similar, even tough this is a new architecture, but perhaps for a conference like ICLR a higher level of distinction of novelty is required in comparison with other methods.

**Questions:**

My question is repeated again:  When does this fail under noise?  It needs to hold up for at least a modest amount in order for it to be relevant.

---

> ### Author Response · Authors · 2025-11-24
> **Response to Reviewer z3pP**
>
> We sincerely appreciate your recognition of our motivation and methods. Below, we will try our best to address your concerns.
>
> **Weaknesses**
>
> > There seems to be no noise study on the method a the method itself really does have to be evaluated on this idea. More broadly, there are works that are pretty similar, even tough this is a new architecture, but perhaps for a conference like ICLR a higher level of distinction of novelty is required in comparison with other methods.
>
> As mentioned in Appendix D, all data generation incorporates random noise. In addition, we have conducted robustness experiments on dataset noise and symmetry noise in Appendices F.1 and F.2, respectively. The results show that DI-SINDy maintains a high identification success rate and prediction accuracy even when the standard SINDy method fails completely, demonstrating its strong robustness against real-world data interference. We also find that DI-SINDy exhibits high tolerance to deviations in the infinitesimal generator, outperforming symmetry-agnostic SINDy when relative perturbation amplitude $\delta \leq 3 \times 10^{-1}$, which highlights its potential for integration with symmetry discovery methods. For more experimental settings and result analyses, please refer to Appendices F.1 and F.2.
>
> In the revised manuscript, we have added Section 5.3, which provides a brief summary of the noise robustness analysis results and directs readers to Appendices F.1 and F.2 for clearer guidance.
>
> **Questions**
>
> > My question is repeated again: When does this fail under noise? It needs to hold up for at least a modest amount in order for it to be relevant.
>
> Please refer to the response to Weakness.
>
> Once again, we sincerely appreciate your valuable feedback, which will help us improve the quality of our paper. We look forward to your response and further discussions ahead!

---

> ### Author Response · Authors · 2025-12-01
> **Rebuttal Summary for AC**
>
> The only concern raised by Reviewer z3pP is regarding noise robustness, which has already been addressed in **Appendices F.1 and F.2** of the initially submitted version. **A simple reminder should be sufficient to clear up this misunderstanding.**

---

### Official Review · Reviewer_cFhz · 2025-10-31

**Soundness:** 2
**Presentation:** 2
**Contribution:** 2
**Rating:** 2
**Confidence:** 4

**Summary:**

The paper proposes a partial differential equation(PDE) discovery process based on differential invariants, aiming at expressing the equation term search space without losses through a symmetrical prior. The main idea is to use infinitesimal generators of the symmetry group to calculate the prolonged term and differential invariants as the skeleton of the equation terms. With the differential invariants basis, the algorithm takes advantage of SINDy to accomplish sparse regression. The DI-SINDy proposed by the authors has shown advantages of a high success rate, strong robustness, long-term forecast stability through several numerical experiments, such as KdV, KS, Burgers, and nKdV PDEs.

**Strengths:**

1. The algorithm systematically introduces the theory of differential invariants into data-driven equation discovery, demonstrating significant theoretical depth and methodological innovation.
2. The method is built upon a solid mathematical foundation, ensuring strict adherence to symmetry constraints.
3. It features a plug-and-play nature that allows for integration with existing discovery methods, offering strong practicality and exhibiting strong robustness against noise and symmetry deviations.

**Weaknesses:**

1. The method's performance is highly dependent on the accuracy of the symmetry information. For complex systems where the symmetries are unknown or difficult to discover, the practical applicability of the method may be limited.
2. The discussion is confined to Lie point symmetries. Compared to a broader perspective on symmetry in machine learning (e.g., as explored in works like "Machine-learning hidden symmetries" by Ziming Liu and Max Tegmark, which deals with Translation invariance, Lie invariance, equivariance, Hamiltonian structure, etc.), the scope of symmetries considered here is narrow.
3. The experimental comparisons are primarily conducted against SINDy-based models. A more comprehensive evaluation is needed, including comparisons with variants of Physics-Informed Neural Networks (PINNs) and other neural-based symbolic regression methods.
4. The DI-SINDy approach assumes that the final governing equation can be expressed as a linear combination of the identified differential invariants. This may limit its expressiveness and generalization ability for systems governed by more complex, non-linear relationships between the invariants. There is a lack of theoretical justification for this linearity assumption, and supporting experiments on equations requiring intricate non-linear combinations of invariants are missing.
5. The paper contains insufficient machine learning content and is recommended for submission to mathematics-focused conferences/journals specializing in PDE research.

**Questions:**

1. Could the framework be extended to incorporate a broader range of symmetries beyond Lie point symmetries, such as those explored in works like "Machine-learning hidden symmetries" (e.g., Hamiltonian structure, gauge invariance)?
2. How does the proposed method compare against other AI-driven paradigms for equation discovery, such as various Physics-Informed Neural Network (PINN) variants or other neural symbolic regression approaches?
3. Can the method's effectiveness be demonstrated on PDEs with stronger, less linear-like non-linearity, or be generalized to the inference of equations in higher-dimensional, coupled systems?
4. Have the authors considered modernizing the core SINDy module itself, for instance, by replacing it with a more expressive symbolic regression model like a Transformer or genetic programming to discover complex combinations of the differential invariants?

---

> ### Author Response · Authors · 2025-11-24
> **Response to Reviewer cFhz (1/3)**
>
> We sincerely appreciate your recognition of the method's innovativeness and solidness. Below, we will try our best to address your concerns point by point.
>
> **Weaknesses**
>
> > W1: The method's performance is highly dependent on the accuracy of the symmetry information. For complex systems where the symmetries are unknown or difficult to discover, the practical applicability of the method may be limited.
>
> For scenarios where the symmetry is unknown, our work can be integrated with existing symmetry discovery methods. In fact, taking the high-dimensional and complex reaction-diffusion system as an example (Section 5.4 and Appendix F.3), we have demonstrated a complete pipeline starting from unknown symmetry, through symmetry discovery, to equation discovery. First, all infinitesimal generators in this case are derived from the results of [1] (an explicit, nonlinear symmetry discovery method), as described in lines 1083-1085 of Appendix F.3 of our paper—this is the first step of the pipeline, discovering symmetries. Although this step is beyond the scope of our paper, the original work [1] provides detailed procedures for obtaining these infinitesimal generators. Subsequently, as demonstrated in the main body of Appendix F.3, we compress the search space based on the symmetries—this is the second step of the pipeline. Combining our work with [1] forms an end-to-end framework for efficiently and accurately discovering governing PDEs from systems with initially unknown symmetries at the current stage.
>
> We acknowledge that symmetry discovery results may be inaccurate, and thus we have also investigated the robustness of our method to noise in the infinitesimal generators (Section 5.3 and Appendix F.2). The conclusion is that, as long as the relative error in the symmetry information does not exceed 30% (i.e., relative perturbation amplitude $\delta \leq 3 \times 10^{-3}$), which is a very high tolerance, the introduction of differential invariants remains beneficial, as discussed in lines 1062-1067 of Appendix F.2.
>
> > W2: The discussion is confined to Lie point symmetries. Compared to a broader perspective on symmetry in machine learning (e.g., as explored in works like "Machine-learning hidden symmetries" by Ziming Liu and Max Tegmark, which deals with Translation invariance, Lie invariance, equivariance, Hamiltonian structure, etc.), the scope of symmetries considered here is narrow.
>
> Lie point symmetry is a very general concept in PDEs. As long as a group transformation can be expressed in the form of an action on points, i.e., $(\tilde{x}, \tilde{u}) = g \cdot (x, u)$, it falls under the category of Lie point symmetries (as described in Section 3, lines 155-161). To our knowledge, almost all research on PDE symmetries revolves around this concept. [2] is an excellent work that discusses different types of symmetries in PDEs, but all of these symmetries are subsets of the concept of Lie point symmetries. Table 1 in [2] provides the linear operators (i.e., infinitesimal generators) of these symmetries, which can serve as inputs for Algorithm 1 in our paper. Then, all of them can be addressed by the pipeline of our work.
>
> > W3: The experimental comparisons are primarily conducted against SINDy-based models. A more comprehensive evaluation is needed, including comparisons with variants of Physics-Informed Neural Networks (PINNs) and other neural-based symbolic regression methods.
>
> In Appendix F.4, we supplement the experiments by combining differential invariants with the Transformer-based symbolic regression method (E2E) [3] and proposing the DI-E2E method. All PDEs in our paper (KdV, KS, Burgers, nKdV, and reaction-diffusion equations) are tested. The results demonstrate that, compared to the original E2E method, our DI-E2E achieves significant performance improvements across all PDEs. It not only exhibits higher prediction accuracy and better stability but also requires shorter inference time. Particularly in complex high-dimensional reaction-diffusion systems, our DI-E2E achieves an almost perfect fit ($R^2 \approx 1$), whereas the original E2E method fails completely. We have added this content in Section 5.5 and refer readers to Appendix F.4 for detailed experimental settings, metric definitions, and result analysis.

---

> ### Author Response · Authors · 2025-11-24
> **Response to Reviewer cFhz (2/3)**
>
> > W4: The DI-SINDy approach assumes that the final governing equation can be expressed as a linear combination of the identified differential invariants. This may limit its expressiveness and generalization ability for systems governed by more complex, non-linear relationships between the invariants. There is a lack of theoretical justification for this linearity assumption, and supporting experiments on equations requiring intricate non-linear combinations of invariants are missing.
>
> The assumption that the equation skeleton can only be a linear combination is an inherent limitation of SINDy, and our contribution focuses on providing differential invariants based on symmetry for existing equation discovery methods. Therefore, this weakness falls outside the scope of this paper. In fact, when our framework is integrated with other types of equation discovery methods (such as transformer-based methods, see response to W3), this limitation naturally disappears. Nonetheless, our DI-SINDy also mitigates this weakness of SINDy to some extent, as symmetry can present complex nonlinear terms in the form of differential invariants (for example, the $e^{-\frac{t}{t_0}} u_t$ term in the nKdV equation). Furthermore, we have already evaluated our approach on high-dimensional complex reaction-diffusion systems (see Section 5.4 and Appendix F.3), which do not follow a simple linear combination of differential invariants.
>
> > W5: The paper contains insufficient machine learning content and is recommended for submission to mathematics-focused conferences/journals specializing in PDE research.
>
> PDE discovery, or symbolic regression, is widely regarded as a classical machine learning task. Its goal is to automatically uncover the underlying mathematical expression structures from discrete trajectory data points of dynamical systems, which fully aligns with the core definition of machine learning—"learning models from data." We suspect you may be trying to point out that our work lacks connection to "black-box" models or deep networks (by the way, we have also supplemented integration with transformer-based methods; see our response to W3). However, not all machine learning involves "black-box" models or deep networks—for instance, traditional machine learning algorithms like decision trees and k-nearest neighbors (k-NN). Moreover, EquivSINDy [4], a similar research work emphasized for comparison in our paper, was also published at NeurIPS, a top machine learning conference of the same category. In summary, we believe the topic of this paper is suitable for ICLR.
>
> **Questions**
>
> > Q1: Could the framework be extended to incorporate a broader range of symmetries beyond Lie point symmetries, such as those explored in works like "Machine-learning hidden symmetries" (e.g., Hamiltonian structure, gauge invariance)?
>
> Please refer to the response to W2.
>
> > Q2: How does the proposed method compare against other AI-driven paradigms for equation discovery, such as various Physics-Informed Neural Network (PINN) variants or other neural symbolic regression approaches?
>
> Please refer to the response to W3.
>
> > Q3: Can the method's effectiveness be demonstrated on PDEs with stronger, less linear-like non-linearity, or be generalized to the inference of equations in higher-dimensional, coupled systems?
>
> We have already evaluated our method's performance on high-dimensional complex reaction-diffusion systems in Appendix F.3. To make it clearer for readers, we add Section 5.4 in the main text to provide a brief summary of the results:
>
> ```quote
> In the high-dimensional case study of reaction-diffusion systems, we validate the capability of the DI-SINDy method in discovering complex partial differential equations. By leveraging differential invariant theory, the method compresses the original search space from hundreds of terms to just 9 terms composed of 5 key invariants, constructing a concise equation skeleton. Experimental results demonstrate that DI-SINDy accurately uncovers the governing equations of the system, including the correct structures of nonlinear reaction and diffusion terms, proving its effectiveness in handling high-dimensional dynamical systems. In contrast, traditional SINDy and EquivSINDy-r methods fail due to the curse of dimensionality. Detailed experimental setups, theoretical derivations, and result analyses are provided in Appendix F.3.
> ```
>
> Additionally, for the newly introduced DI-E2E, we also conduct tests on reaction-diffusion systems. The results show that while traditional E2E completely fails for such nonlinear coupled systems, our DI-E2E achieves a near-perfect $R^2$-score (close to 1). Further details can be found in Section 5.5 and Appendix F.4.

---

> ### Author Response · Authors · 2025-11-24
> **Response to Reviewer cFhz (3/3)**
>
> > Q4: Have the authors considered modernizing the core SINDy module itself, for instance, by replacing it with a more expressive symbolic regression model like a Transformer or genetic programming to discover complex combinations of the differential invariants?
>
> Please refer to the response to W3.
>
> Once again, we sincerely appreciate your valuable feedback, which will help us improve the quality of our paper. We look forward to your response and further discussions ahead!
>
> **References**
>
> [1] Hu, Lexiang, Yikang Li, and Zhouchen Lin. "Explicit discovery of nonlinear symmetries from dynamic data." ICML 2025.
>
> [2] Liu, Ziming, and Max Tegmark. "Machine learning hidden symmetries." Physical Review Letters 128, no. 18 (2022): 180201.
>
> [3] Kamienny, Pierre-Alexandre, Stéphane d'Ascoli, Guillaume Lample, and François Charton. "End-to-end symbolic regression with transformers." NeurIPS 2022.
>
> [4] Yang, Jianke, Wang Rao, Nima Dehmamy, Robin Walters, and Rose Yu. "Symmetry-informed governing equation discovery." NeurIPS 2024.

---

> ### Author Response · Authors · 2025-12-01
> **Rebuttal Summary for AC**
>
> Reviewer cFhz acknowledged our paper for its **significant theoretical depth and methodological innovation**, **solid mathematical foundation**, **strong practicality**, and **strong robustness**.
>
> Our responses to the concerns are briefly summarized below.
>
> W1: The symmetries provided in **Appendix F.3** of the initially submitted version are derived from LieNLSD [1] rather than prior knowledge. The impact of symmetry discovery errors has already been studied in **Appendix F.2** of the initially submitted version.
>
> W2: Lie point symmetries are very general concepts in PDEs. The symmetries mentioned by Reviewer cFhz are **all included within the scope of Lie point symmetries**.
>
> W3: We have added **Appendix F.4** in the revised version.
>
> W4: This is a **limitation of SINDy rather than our pipeline**. Applying our pipeline to other equation discovery methods can **naturally eliminate this limitation**, as shown in Appendix F.4. Cases involving nonlinear terms have been included in **Appendix F.3** of the initially submitted version.
>
> W5: **Completely incorrect viewpoint**. Our responses in the rebuttal and Reviewer VGyg's comment that our work is "**both rigorous and well-integrated into the machine learning context**" can easily refute this.
>
> Q1: Please refer to W2.
>
> Q2: Please refer to W3.
>
> Q3: Already included in **Appendix F.3** of the initially submitted version.
>
> Q4: Please refer to W3.
>
> In summary, most of Reviewer cFhz’s concerns stem from **overlooking the Appendix and misunderstandings of related concepts**. We believe that if Reviewer cFhz objectively and responsibly reads our rebuttal, his/her opinion will **easily turn positive**.
>
> **References**
>
> [1] Hu, Lexiang, Yikang Li, and Zhouchen Lin. "Explicit discovery of nonlinear symmetries from dynamic data." ICML 2025.

---

### Official Review · Reviewer_VGyg · 2025-10-31

**Soundness:** 3
**Presentation:** 3
**Contribution:** 4
**Rating:** 6
**Confidence:** 4

**Summary:**

This paper introduces a symmetry-guided framework for governing equation discovery based on differential invariants. The key idea is to embed known (or discovered) symmetries into the equation discovery process by constructing invariant feature libraries derived from the symmetry group’s infinitesimal generators. This approach effectively reduces the search space without sacrificing expressiveness or correctness. The authors formalize this through a completeness proposition and implement the framework within SINDy (resulting in DI-SINDy). Empirical results on several canonical PDEs (e.g., KdV, Burgers, KS, nKdV) show improved identification accuracy, success rates, and long-term prediction stability compared to baseline methods such as SINDy and EquivSINDy.

**Strengths:**

This paper offers a theoretically sound and elegant contribution to the field of equation discovery. By leveraging differential invariants derived from known or learned symmetries, the authors present a clear and general framework that can be applied to improve many existing methods such as SINDy. The theoretical foundation, grounded in classical Lie symmetry analysis, is both rigorous and well-integrated into the machine learning context.
The conceptual innovation of embedding symmetry directly into the function library through invariant construction represents a meaningful advance over prior symmetry-regularized approaches. Importantly, the proposition ensuring lossless expressivity gives the method strong theoretical credibility. From a presentation standpoint, the paper is well-written and visually clear. The introduction and related work are accessible, providing an intuitive motivation for readers from both applied mathematics and ML backgrounds. Figures, particularly the main pipeline diagram, effectively communicate the overall workflow. The results section is also strong: quantitative tables and rollout plots convincingly demonstrate that DI-SINDy outperforms baselines across multiple PDE benchmarks.

**Weaknesses:**

Despite its theoretical depth, the paper’s empirical scope is somewhat narrow. The experiments are limited to one-dimensional canonical PDEs such as Burgers, KdV, and KS, leaving open questions about scalability to higher-dimensional or real-world systems. The claim of computational efficiency is also not empirically validated, while the method does reduce the search space, the additional overhead of computing differential invariants is not benchmarked. A runtime or complexity analysis would help substantiate this claim.
Furthermore, the mathematical exposition can be dense, particularly in the sections involving infinitesimal generators and prolongations. Readers without a background in Lie group theory may find these derivations difficult to follow. Including a small, concrete example would greatly improve accessibility. Finally, the presentation of results tables could be clearer: key tables are introduced without sufficient context, requiring readers to read ahead before fully understanding the metrics. Overall, while the theoretical foundation is excellent, the paper would benefit from more diverse experiments, clearer runtime validation, and slightly improved accessibility in the methodological sections.

**Questions:**

1) How does DI-SINDy perform when the input symmetries are incomplete or slightly mis-specified?

2) Have you evaluated the runtime trade-offs between computing differential invariants and the reduction in search-space size?

3) Can this framework be extended to approximate or data-driven symmetries, where invariance holds only approximately?

4) How does noise in the symmetry generators or dataset affect performance, and can robustness results from the appendix be summarized in the main text?

---

> ### Author Response · Authors · 2025-11-24
> **Response to Reviewer VGyg (1/2)**
>
> We sincerely appreciate your comprehensive recognition of the innovation, contribution, methodological soundness, presentation, and experimental results! Below, we will try our best to address your concerns point by point.
>
> **Weaknesses**
>
> > W1: Despite its theoretical depth, the paper’s empirical scope is somewhat narrow. The experiments are limited to one-dimensional canonical PDEs such as Burgers, KdV, and KS, leaving open questions about scalability to higher-dimensional or real-world systems.
>
> We have already evaluated our method's performance on high-dimensional complex reaction-diffusion systems in Appendix F.3. To make it clearer for readers, we add Section 5.4 in the main text to provide a brief summary of the results:
>
> ```quote
> In the high-dimensional case study of reaction-diffusion systems, we validate the capability of the DI-SINDy method in discovering complex partial differential equations. By leveraging differential invariant theory, the method compresses the original search space from hundreds of terms to just 9 terms composed of 5 key invariants, constructing a concise equation skeleton. Experimental results demonstrate that DI-SINDy accurately uncovers the governing equations of the system, including the correct structures of nonlinear reaction and diffusion terms, proving its effectiveness in handling high-dimensional dynamical systems. In contrast, traditional SINDy and EquivSINDy-r methods fail due to the curse of dimensionality. Detailed experimental setups, theoretical derivations, and result analyses are provided in Appendix F.3.
> ```
>
> Additionally, for the newly introduced DI-E2E, we also conduct tests on reaction-diffusion systems. The results show that while traditional E2E completely fails for such nonlinear coupled systems, our DI-E2E achieves a near-perfect $R^2$-score (close to 1). Further details can be found in Section 5.5 and Appendix F.4.
>
> > W2: The claim of computational efficiency is also not empirically validated, while the method does reduce the search space, the additional overhead of computing differential invariants is not benchmarked. A runtime or complexity analysis would help substantiate this claim.
>
> In Appendix F.4, We supplement the experiments by combining differential invariants with the Transformer-based symbolic regression method (E2E) [1] and proposing the DI-E2E method. All PDEs in our paper (KdV, KS, Burgers, nKdV, and reaction-diffusion equations) are tested. As shown in Table 8, compared to the original E2E method, our DI-E2E significantly reduces time overhead. In particular, for the reaction-diffusion system, even when we sample only 8,192 data points for efficient inference, our DI-E2E saves approximately 15 seconds of inference time compared to E2E. As the amount of data increases, this gap will continue to widen. On the other hand, our tests show that using LieNLSD [2] to discover infinitesimal generators from the reaction-diffusion system takes only 3–4 seconds. Clearly, this complete pipeline—from unknown symmetries, to discovering symmetries, to symmetry-guided equation discovery—not only achieves higher accuracy and adheres to physical laws but also significantly reduces time consumption compared to direct symbolic regression from the original search space.
>
> For more details regarding the experimental setup, definition of evaluation metrics, and analysis of results for E2E and DI-E2E, please refer to Appendix F.4. In Section 5.5, we briefly summarize the main experimental conclusions and direct readers to Appendix F.4 for further clarity.
>
> > W3: Furthermore, the mathematical exposition can be dense, particularly in the sections involving infinitesimal generators and prolongations. Readers without a background in Lie group theory may find these derivations difficult to follow. Including a small, concrete example would greatly improve accessibility.
>
> We believe that a non-expert audience tends to find two aspects of the theoretical framework confusing: the related concepts of Lie point symmetry and the derivation process of differential invariants. Accordingly, we have provided a concrete example of Lie point symmetry in Appendix B and demonstrated the detailed derivation process of differential invariants in Appendix C to help readers gain an intuitive understanding. In the revised manuscript, we also add detailed references to them at the beginning of Section 3: Preliminary and Section 4: Method (lines 138–141 and lines 184–187), respectively, with bold highlighting. We are confident this will significantly improve the readability of the paper.

---

> ### Author Response · Authors · 2025-11-24
> **Response to Reviewer VGyg (2/2)**
>
> > W4: Finally, the presentation of results tables could be clearer: key tables are introduced without sufficient context, requiring readers to read ahead before fully understanding the metrics.
>
> We have double-checked that the metrics in Table 3 (the only experimental results table in the main text) were clearly defined (lines 393–402) before the reference was made (line 427). Could you specify which table presentation appears confusing? We will respond accordingly based on your further guidance. Thank you for your valuable suggestions!
>
> **Questions**
>
> > Q1: How does DI-SINDy perform when the input symmetries are incomplete or slightly mis-specified?
>
> In Appendix F.2, we have investigated the robustness of DI-SINDy to noise in the infinitesimal generators. The conclusion is that, as long as the relative error in the symmetry information does not exceed 30% (i.e., relative perturbation amplitude $\delta \leq 3 \times 10^{-3}$), which is a very high tolerance, the introduction of differential invariants remains beneficial, as discussed in lines 1062-1067 of Appendix F.2. In the revised manuscript, we have added Section 5.3, which provides a brief summary of the aforementioned experimental results and directs readers to Appendix F.2 for further details.
>
> > Q2: Have you evaluated the runtime trade-offs between computing differential invariants and the reduction in search-space size?
>
> Please refer to the response to W2.
>
> > Q3: Can this framework be extended to approximate or data-driven symmetries, where invariance holds only approximately?
>
> For approximate or data-driven symmetries, although it is difficult to know them in advance, we can obtain them using symmetry discovery methods. In Appendix F.3, we have already taken the high-dimensional complex reaction-diffusion system as an example to demonstrate a complete pipeline that starts from unknown symmetries, proceeds through symmetry discovery, and culminates in equation discovery. First, all infinitesimal generators in this case are derived from the results of [2] (an explicit, nonlinear symmetry discovery method), as described in lines 1083-1085 of Appendix F.3 of our paper—this is the first step of the pipeline, discovering symmetries. Although this step is beyond the scope of our paper, the original work [2] provides detailed procedures for obtaining these infinitesimal generators. Subsequently, as demonstrated in the main body of Appendix F.3, we compress the search space based on the symmetries—this is the second step of the pipeline. Combining our work with [2] forms an end-to-end framework for efficiently and accurately discovering governing PDEs from systems with initially unknown symmetries at the current stage.
>
> For inaccurate symmetry discovery results, we have also investigated the robustness of our method to noise in the infinitesimal generators in Appendix F.2. Please refer to the response to Q1 for more details.
>
> > Q4: How does noise in the symmetry generators or dataset affect performance, and can robustness results from the appendix be summarized in the main text?
>
> In the revised manuscript, we added Section 5.3 to summarize the results of the noise robustness analysis:
>
> > We test the stability of our DI-SINDy method through robustness experiments under conditions such as dataset noise and inaccurate symmetry priors. First, we progressively increase the noise level ($\sigma = \\{3 \times 10^{-3}, 5 \times 10^{-3}\\}$) on the Burgers equation dataset. The results show that DI-SINDy maintains a high identification success rate and prediction accuracy even when the standard SINDy method fails completely, demonstrating its strong robustness against real-world data interference. Second, we simulate scenarios with deviations in the infinitesimal generator (relative perturbation amplitude $\delta = \\{10^{-1}, 3 \times 10^{-1}, 5 \times 10^{-1}\\}$) and find that DI-SINDy also exhibits high tolerance, outperforming symmetry-agnostic SINDy when $\delta \leq 3 \times 10^{-1}$, highlighting its potential for integration with symmetry discovery methods. Detailed experimental setups and result analyses are provided in Appendices F.1 and F.2.
>
> We believe this change will provide clearer guidance for readers. Thank you for your suggestion!
>
> Once again, we sincerely appreciate your valuable feedback, which will help us improve the quality of our paper. We look forward to your response and further discussions ahead!
>
> **References**
>
> [1] Kamienny, Pierre-Alexandre, Stéphane d'Ascoli, Guillaume Lample, and François Charton. "End-to-end symbolic regression with transformers." NeurIPS 2022.
>
> [2] Hu, Lexiang, Yikang Li, and Zhouchen Lin. "Explicit discovery of nonlinear symmetries from dynamic data." ICML 2025.

---

> > ### Author Response · Authors · 2025-12-01
> > **Rebuttal Summary for AC**
> >
> > Reviewer VGyg has offered high praise for our work, describing it with terms such as "**theoretically sound and elegant contribution**", "**clear and general framework**", "**both rigorous and well-integrated into the machine learning context**", "**meaningful advance over prior symmetry-regularized approaches**", "**strong theoretical credibility**", "**well-written and visually clear**", and "**accessible and intuitive**".
> >
> > Our responses to the concerns are briefly summarized below.
> >
> > W1: Already included in **Appendix F.3** of the initially submitted version.
> >
> > W2: Already reported in the newly added **Appendix F.4** of the revised version.
> >
> > W3: Concrete examples have already been provided in **Appendices B and C** of the initially submitted version.
> >
> > W4: We have double-checked that the metrics are already detailed in the **first paragraph of Section 5.2** of the initially submitted version.
> >
> > Q1: Already included in **Appendix F.2** of the initially submitted version.
> >
> > Q2: Already reported in the newly added **Appendix F.4** of the revised version.
> >
> > Q3: Already included in **Appendix F.2** of the initially submitted version.
> >
> > Q4: Already included in **Appendices F.1 and F.2** of the initially submitted version, with a summary added in **Section 5.3** of the revised version.
> >
> > In conclusion, we are confident that Reviewer VGyg will continue to **champion the acceptance** of this paper.

---

### Author Response · Authors · 2025-11-24
**Revision Summary**

**All modifications are highlighted in blue text.**

- Added instructions for non-specialist readers in the introductions of Section 3: Preliminary and Section 4: Method.

- Corrected a typo in Equation (5).

- Added Appendix A.4 to supplement experiments on combining differential invariants with transformer-based equation discovery methods; accordingly updated the titles and referencing texts of Table 3, Figure 3, Table 5, and Figure 5 to distinguish between SINDy-based methods and transformer-based methods.

- Added Sections 5.3, 5.4, and 5.5 in the main text to briefly summarize the additional experimental results in Appendix F.

---

### Author Response · Authors · 2025-11-30
**Important Message to AC!**

We believe that we are one of the biggest victims of this data leak incident. Although the initial ratings were low, we found that **almost all of the reviewers' concerns stemmed from their oversight of the Appendix in the initially submitted version** (note: not the revised version during the rebuttal), despite corresponding guidance being provided in the main text. According to the standard process, **a simple reminder would have easily cleared up the reviewers' misunderstandings, thereby shifting their opinions positively**. However, all of this was interrupted by the unforeseen incident. Therefore, we sincerely request that you **disregard the bias from the initial ratings** and provide a fair and objective decision. We deeply appreciate your additional hard work and dedication!

---

> ### Author Response · Authors · 2025-11-30
> **Overall Rebuttal Summary for AC**
>
> We have categorized the reviewers' concerns into the following main points:
>
> 1. Robustness analysis of dataset noise (Reviewer VGyg's Q4; Reviewer z3pP's only weakness): Already included in Appendix F.1 of the initially submitted version.
>
> 2. Robustness analysis of symmetric noise (Reviewer VGyg's Q1, Q3, Q4; Reviewer z3pP's only weakness; Reviewer ofA3's W2): Already included in Appendix F.2 of the initially submitted version.
>
> 3. High-dimensional complex systems (Reviewer VGyg's W1; Reviewer cFhz's W4, Q3; Reviewer ofA3's Q2): Already included in Appendix F.3 of the initially submitted version.
>
> 4. Cases of unknown symmetries (Reviewer cFhz's W1; Reviewer ofA3's W2): The symmetries provided in Appendix F.3 of the initially submitted version are derived from LieNLSD [1] rather than prior knowledge. The impact of symmetry discovery errors has already been studied in Appendix F.2 of the initially submitted version.
>
> 5. Differential invariants for more equation discovery methods beyond SINDy (Reviewer cFhz's W3, Q2, Q4): We have added Appendix F.4 in the revised version.
>
> 6. Time cost (Reviewer VGyg's W2, Q2; Reviewer ofA3's Q1): Already reported in Appendix F.4 added in the revised version.
>
> 7. Readability of the theory (Reviewer VGyg's W3; Reviewer ofA3's W1): Concrete examples have already been provided in Appendix B and Appendix C of the initially submitted version.
>
> **References**
>
> [1] Hu, Lexiang, Yikang Li, and Zhouchen Lin. "Explicit discovery of nonlinear symmetries from dynamic data." ICML 2025.

---

### Meta-Review · Area_Chair_tCY1 · 2026-01-06

**Summary:**

This paper proposes a method for estimating partial differential equations from data that leverages the system's inherent symmetries to reduce the search space, thereby improving estimation accuracy and efficiency. The reviewers acknowledge that this is a method with strong theoretical support. However, they also raise concerns about its practical applicability.  In response, the authors argue that experiments addressing symmetry errors and high-dimensional problems are shown in the appendix. While the experiments are certainly in the appendix and some of the reviewers' concerns may have been addressed, I remain skeptical whether they sufficiently resolve all concerns. I believe that significant rewriting is necessary for this paper to be accepted.

**Reviewer Concerns:**

Major concerns include the assumption of symmetry and the limited experiments for high-dimensional problems. Although robustness against symmetry errors has been experimentally investigated, these experiments are limited. Furthermore, performance degrades when the noise level is high. Similarly, the high-dimensional problems addressed are also limited, and it cannot be said that the proposed method demonstrates sufficient practical effectiveness. I believe that these points are considered important for practical applications, and a major revision is necessary, including additional experiments.

**Reviewer Scores:**

The original scores were 6-4-2-2; however, one of the reviewers who gave a score of 2 acknowledged that his/her concerns were addressed. So, the score may become 6-6-4-2. In addition, the other two negative reviewers might raise their scores. If one of them did, it could become 6-6-4-4 or 6-6-6-2. This seems slightly below the required score for acceptance.

---

### Decision · Program_Chairs · 2026-01-26

Reject